# Comparison of Clinical Outcomes after Non-ST-Segment and ST-Segment Elevation Myocardial Infarction in Diabetic and Nondiabetic Populations

**DOI:** 10.3390/jcm11175079

**Published:** 2022-08-29

**Authors:** Yong Hoon Kim, Ae-Young Her, Seung-Woon Rha, Cheol Ung Choi, Byoung Geol Choi, Ji Bak Kim, Soohyung Park, Dong Oh Kang, Ji Young Park, Sang-Ho Park, Myung Ho Jeong

**Affiliations:** 1Division of Cardiology, Department of Internal Medicine, Kangwon National University School of Medicine, Chuncheon 24289, Korea; 2Cardiovascular Center, Korea University Guro Hospital, Seoul 08308, Korea; 3Cardiovascular Research Institute, Korea University College of Medicine, Seoul 02841, Korea; 4Division of Cardiology, Department of Internal Medicine, Cardiovascular Center, Nowon Eulji Medical Center, Eulji University, Seoul 01830, Korea; 5Cardiology Department, Soonchunhyang University Cheonan Hospital, Cheonan 31151, Korea; 6Department of Cardiology, Cardiovascular Center, Chonnam National University Hospital, Gwangju 61469, Korea

**Keywords:** diabetes, non-ST-elevation myocardial infarction, outcomes, ST-elevation myocardial infarction

## Abstract

Using a new-generation drug-eluting stent, we compared the 2-year clinical outcomes of patients with diabetes mellitus (DM) and non-DM concomitant with a non-ST-segment elevation myocardial infarction (NSTEMI) and an ST-segment elevation myocardial infarction (STEMI) who underwent percutaneous coronary intervention. A total of 11,798 patients with acute myocardial infarction were classified into two groups: DM (NSTEMI, *n* = 2399; STEMI, *n* = 2693) and non-DM (NSTEMI, *n* = 2694; STEMI, *n* = 4012). The primary clinical outcome was the occurrence of major adverse cardiac events (MACE), defined as all-cause death, recurrent myocardial infarction, or any coronary repeat revascularization. The secondary outcome was the occurrence of definite or probable stent thrombosis. In all the patients, both multivariable and propensity score-adjusted analyses revealed that the incidence rates of MACE (adjusted hazard ratio (aHR), 1.214; *p* = 0.006 and aHR, 1.298; *p* = 0.002, respectively), all-cause death, cardiac death (CD), and non-CD rate were significantly higher in the NSTEMI group than in the STEMI group. Additionally, among patients with NSTEMI, there was a higher non-CD rate (aHR, 2.200; *p* = 0.007 and aHR, 2.484; *p* = 0.004, respectively) in the DM group and a higher CD rate (aHR, 2.688; *p* < 0.001 and 2.882; *p* < 0.001, respectively) in the non-DM group. In this retrospective study, patients with NSTEMI had a significantly higher 2-year mortality rate than those with STEMI did. Furthermore, strategies to reduce the non-CD rate in patients with DM and the CD rate in patients without DM could be beneficial for those with NSTEMI.

## 1. Introduction

Diabetes mellitus (DM), particularly type 2 DM (T2DM), is one of the most important threats to public health in the twenty-first century [1]. In the Harmonizing Outcomes with RevasculariZatiON and Stents in Acute Myocardial Infarction (AMI) trial [2], patients with DM and those with newly diagnosed DM had higher 3-year death rates than non-DM patients (11%, 12%, and 6%, respectively). Thrombus formation after rupture or erosion of vulnerable atherosclerotic plaques is a common pathophysiology in both non-ST-segment elevation myocardial infarction (NSTEMI) and ST-segment elevation myocardial infarction (STEMI) [3]. However, patients with NSTEMI have partial or intermittent occlusion of the coronary artery, whereas patients with STEMI often have complete occlusion [4]. Additionally, after a certain duration of complete ischemia, there are no interventions that can salvage the ischemic myocardium [5]. Because cardiogenic shock complicates an increasing number of STEMI cases [6], Polonski et al. suggested that in-hospital outcomes were worse in patients with STEMI than in those with NSTEMI. Patients with NSTEMI have a greater prevalence of comorbidities [7] and have received relatively fewer guideline-based treatments than patients with STEMI [8]. Hence, long-term mortality is higher in patients with NSTEMI than in those with STEMI [9,10]. Until now, comparative results between NSTEMI and STEMI have been conflicting, and further discussion is needed [10,11]. Because recent reports concerning long-term clinical outcomes in patients with and without DM are confined to patients with NSTEMI [12] or STEMI [13], data on head-to-head comparisons between long-term clinical outcomes in NSTEMI and STEMI in patients with and without DM are scarce. Currently, new-generation drug-eluting stents (DESs) have nearly replaced bare-metal stents (BMSs) and first-generation DESs (1G-DESs) for routine percutaneous coronary intervention (PCI) [14]. New-generation DESs are more effective than 1G-DESs in reducing major clinical outcomes in patients with DM [14]. Furthermore, to the best of our knowledge, no specific large-scale study has compared the long-term clinical outcomes between the NSTEMI and STEMI groups in patients with and without DM after PCI using new-generation DESs to reflect current real-world practice. Hence, this study, using a new-generation DES, evaluated the 2-year comparative clinical outcomes between two different types of AMI (NSTEMI versus STEMI) in patients with DM and non-DM who underwent successful PCI.

## 2. Methods

### 2.1. Study Population

This non-randomized multicenter observational retrospective cohort study enrolled 21,343 patients with AMI aged ≥30 years at the onset of DM who underwent successful PCI with newer-generation DESs between November 2005 and June 2015; data was obtained from the Korea AMI Registry (KAMIR). To ensure that only individuals with T2DM were included, patients aged <30 years at the onset of DM were excluded based on a previous study [15]. KAMIR was established in November 2005 and involves more than 50 communities and teaching hospitals in South Korea [16]. Patients with the following conditions were excluded: incomplete laboratory results including unidentified results of blood hemoglobin (Hb) A1c and blood glucose (*n* = 8314; 39.0%), lost to follow-up (*n* = 1067; 5.0%), and in-hospital death (*n* = 164; 0.8%). After exclusion, 11,798 patients with AMI who underwent successful PCI using new-generation DESs were included. The patients were classified into DM (*n* = 5092; 43.2%) and non-DM (*n* = 6706; 56.8%) groups (Figure 1). Thereafter, these two groups were further sub-classified into NSTEMI (group A, *n* = 2399 (47.1%) and group C, *n* = 2694 (40.2%), respectively) and STEMI (group B, *n* = 2693 (52.9%) and group D, *n* = 4012 (59.8%), respectively). This study was conducted in accordance with the ethical guidelines of the 2004 Declaration of Helsinki and was approved by the ethics committee of each participating center and the Chonnam National University Hospital Institutional Review Board ethics committee (CNUH-2011-172). All 11,798 patients included in the study provided written informed consent prior to enrollment and completed a 2-year clinical follow-up through face-to-face interviews, phone calls, or chart reviews. An independent event adjudication committee evaluated all the clinical events. The event adjudication process has been previously described by KAMIR investigators [16].

### 2.2. Percutaneous Coronary Intervention and Medical Treatment

Diagnostic coronary angiography and PCI were performed according to the general guidelines [17]. Loading doses of aspirin (200–300 mg), clopidogrel (300–600 mg), ticagrelor (180 mg), and prasugrel (60 mg) were administered to all the enrolled patients prior to PCI. Subsequently, dual antiplatelet therapy (a combination of aspirin (100 mg/day) with clopidogrel (75 mg/day), or ticagrelor (90 mg twice a day), or prasugrel (5–10 mg/day)) was recommended for >12 months. Based on previous reports [18,19], triple antiplatelet therapy was administered (100 mg of cilostazol was administered twice a day in addition to the dual antiplatelet therapy) at the discretion of the individual operator. Moreover, the access site, revascularization strategy, and DES selection were assigned to the individual operators and were to be performed at their own discretion.

### 2.3. Study Definitions and Clinical Outcomes

The DM group included patients with HbA1c, fasting plasma glucose, and/or random plasma glucose levels of ≥6.5%, ≥126 mg/dL (7.0 mmol/L), and ≥200 mg/dL (11.1 mmol/L) at index hospitalization, respectively, according to the American Association’s clinical practice recommendations [20]. In addition to their medical history, patients with known diabetes, for which they received medical treatment (insulin or antidiabetic), or newly diagnosed diabetes were also included in the DM group. NSTEMI was defined as the absence of persistent ST-segment elevation with increased levels of cardiac biomarkers and the appropriate clinical context [21,22]. STEMI was defined as follows: ongoing chest pain and electrocardiogram findings on admission showing an ST-segment elevation in at least two contiguous leads of ≥2 mm (0.2 mV) in men or ≥1.5 mm (0.15 mV) in women in leads V2–V3 and/or ≥1 mm (0.1 mV) in other contiguous chest leads, limb leads, or new-onset left bundle branch block [23,24]. The primary clinical outcome of this study was the occurrence of major adverse cardiac events (MACE), defined as all-cause death, recurrent myocardial infarction (re-MI), or repeat coronary revascularization, including target lesion revascularization, target vessel revascularization (TVR), and non-TVR. The secondary outcome was definite or probable stent thrombosis during the 2-year follow-up period. All-cause death was considered as cardiac death (CD) unless an undisputed non-cardiac cause was present [25]. The definitions of re-MI, target lesion revascularization, TVR, and non-TVR have previously been published [26]. The types of new-generation DESs used are listed in Table 1. Glomerular function was calculated using the Chronic Kidney Disease Epidemiology Collaboration equation for the estimated glomerular filtration rate (eGFR) [27]. In our study, patients who required multivessel PCI included those who underwent PCI of the non-infarct-related artery (IRA) during index PCI of the IRA or who underwent staged PCI for the non-IRA within the index hospitalization. Therefore, patients who underwent staged PCI after discharge were excluded from this study because reperfusion timing could have acted as a bias.

### 2.4. Statistical Analyses

For continuous variables, between-group differences were evaluated using unpaired *t*-tests. Data are expressed as mean ± standard deviation. For discrete variables, between-group differences were expressed as counts and percentages and analyzed using the chi-squared or Fisher’s exact test. We tested all variables with a *p* value of <0.001 in the univariate analysis between the NSTEMI and STEMI groups. After the univariate analysis, we performed a multicollinearity test [28] between the included variables to confirm that there was no definite collinearity between them (Appendix A). The variance inflation factor values were calculated to measure the degree of multicollinearity among the variables. A variance inflation factor of >5 indicates a high correlation [29]. Multicollinearity was considered when the tolerance value was <0.1 [30] or the condition index was >10 [29]. The variables included in the multivariable Cox regression analysis were as follows: male sex, age, left ventricular ejection fraction (LVEF), body mass index, systolic blood pressure, diastolic blood pressure, cardiogenic shock, Killip class I/II, cardiopulmonary resuscitation (CPR) on admission, hypertension, dyslipidemia, previous myocardial infarction, previous PCI, previous coronary artery bypass graft, previous heart failure, previous cerebrovascular accidents (CVA), current smoker, peak creatine kinase myocardial band (CK-MB), peak troponin-I, N-terminal pro-brain natriuretic peptide (NT-ProBNP), serum creatinine, eGFR, total cholesterol, triglyceride, low-density lipoprotein cholesterol, clopidogrel, ticagrelor, prasugrel, angiotensin-converting enzyme inhibitor (ACEI), angiotensin receptor blocker (ARB), beta-blocker, calcium channel blocker, and lipid-lowering agent. Moreover, to adjust for potential confounders, a propensity score (PS)-adjusted analysis was performed using a logistic regression model. We tested all the available variables that could be of potential relevance, including baseline clinical, angiographic, and procedural factors (Table 1). The c-statistic for the PS-matched analysis in this study was 0.712. Patients in the NSTEMI group were matched to those in the STEMI group (1:1) according to PSs using the nearest available pair-matching method. The patients were matched using a caliper width of 0.01. This procedure yielded 5,768 well-matched pairs (Appendix A). Various clinical outcomes were estimated using the Kaplan–Meier curve analysis, and group differences were compared using the log-rank test. Statistical significance was defined as a 2-tailed *p* value of <0.05. All the statistical analyses were performed using the SPSS software version 20 (IBM, Armonk, NY, USA).

## 3. Results

### 3.1. Baseline Characteristics

Table 1 and Appendix A show the baseline, laboratory, angiographic, and procedural characteristics of the study population. In both the DM and non-DM groups, patients in the NSTEMI group had a higher mean age, mean values of LVEF, systolic blood pressure, diastolic blood pressure, NT-ProBNP levels, diameter of the deployed stent, length of deployed stent, and mean number of deployed stents than patients in the STEMI group. The numbers of the following patients were higher in the NSTEMI group: with hypertension; dyslipidemia; previous histories of PCI and CVA; prescribed with ARB, left main coronary artery and left circumflex artery as the IRA and treated vessels; American College of Cardiology/American Heart Association type B2 lesions and multivessel diseases; and those who received transradial approach and underwent intravascular ultrasound examination. The STEMI group included the following patients: a higher number of men; cardiogenic shock; underwent CPR on admission; current smokers; prescribed with ACEI and glycoprotein II/IIIa; right coronary artery as an IRA and treated vessel; American College of Cardiology/American Heart Association type C lesions; single-vessel disease and pre-PCI thrombolysis in myocardial infarction (TIMI) flow grade 0/1; and those who underwent PCI within 24 h compared to those in the NSTEMI group. In both the NSTEMI and STEMI groups, patients in the DM group had a higher mean age, mean body mass index values, NT-ProBNP levels, triglyceride levels, and mean number of deployed stents than those in the non-DM group (Appendix A). The number of patients with hypertension; dyslipidemia; previous history of MI, PCI, coronary artery bypass graft, and CVA; those prescribed ARB; and those with RCA as a treated vessel and multivessel disease were also higher in the DM group. The non-DM group included the following patients: a higher number of men; underwent CPR on admission; current smokers; those prescribed with ticagrelor, ACEI, lipid-lowering agent, and glycoprotein II/IIIa; and those with single-vessel disease compared to those in the DM group. The mean values of LVEF, peak CK-MB, total cholesterol, low-density lipoprotein cholesterol, and number of patients with pre-PCI TIMI flow grade 0/1 were also higher in the non-DM group.

### 3.2. Clinical Outcomes

The cumulative incidences of major clinical outcomes during the 2-year follow-up period are summarized in Table 2 and Table 3 and Figure 2. In the DM group, after a multivariable-adjusted analysis, the incidence rates of MACE (adjusted hazard ratio (aHR), 1.098; 95% confidence interval [CI], 0.875–1.314; *p* = 0.401), all-cause death, CD, re-MI, any repeat revascularization, and ST were not significantly different between the NSTEMI and STEMI groups (Table 2). However, the non-CD rate was significantly higher in the NSTEMI group than that in the STEMI group (aHR, 2.200; 95% CI, 1.231–3.813; *p* = 0.007). After a PS-adjusted analysis, the non-CD rate was significantly higher in the NSTEMI group than in the STEMI group (aHR, 2.484; 95% CI, 1.326–4.651; *p* = 0.004). In the non-DM group, after a multivariable-adjusted analysis, the rates of MACE (aHR, 1.384; *p* = 0.002), all-cause death (aHR, 2.054; *p* < 0.01), and CD (aHR, 2.688; 95% CI, *p* < 0.001) were significantly higher in the NSTEMI group than in the STEMI group (Table 2). After a PS-adjusted analysis, the rates of MACE (aHR, 1.543; *p* < 0.001), all-cause death (aHR, 2.172; *p* < 0.001), and CD (aHR, 2.882; *p* < 0.001) were also higher in the NSTEMI group than in the STEMI group. In all the patients, after both multivariable- and PS-adjusted analyses, the rates of MACE (aHR, 1.214; *p* = 0.006 and aHR, 1.298; *p* = 0.002, respectively), all-cause death (aHR, 1.521; *p* < 0.001 and aHR, 1.653; *p* < 0.001, respectively), CD (aHR, 1.367; *p* = 0.041 and aHR, 1.499; *p* = 0.022, respectively), and non-CD rate (aHR, 1.745; *p* = 0.005 and aHR, 1.977; *p* = 0.004, respectively) were significantly higher in the NSTEMI group than in the STEMI group. Table 3 compares the clinical outcomes of the DM and non-DM groups. In the NSTEMI group, after a multivariable-adjusted analysis, the rates of MACE (aHR, 1.326; *p* = 0.007), all-cause death (aHR, 1.701; *p* = 0.001), non-CD rate (aHR, 2.549; *p* < 0.001), and ST (aHR, 2.272; *p* = 0.048) were significantly higher in the DM group than in the non-DM group. However, the CD rates were similar between the two groups. In the STEMI group, the rates of MACE (aHR, 1.481; *p* < 0.001), all-cause death (aHR, 1.869; *p* < 0.001), CD (aHR, 2.248; *p* < 0.001), re-MI (aHR, 1.537; *p* = 0.023), and any repeat revascularization (aHR, 1.374; *p* = 0.024) were significantly higher in the DM group than in the non-DM group. However, the non-CD rates were similar between the two groups. Overall, all of the clinical outcomes were worse in the DM group than those in the non-DM group. Appendix A shows the causes of non-CD in this study. In the DM group, the multiple organ failure (0.7% vs. 0.2%, *p* = 0.008) and CVA (0.9% vs. 0.4%, *p* = 0.034) rates were significantly higher in the NSTEMI group than in the STEMI group. However, in the non-DM group, none of the causes of non-CD, including multiple organ failure and CVA, were significantly different between the NSTEMI and STEMI groups. Appendix A shows the independent predictors of MACE. Reduced LVEF (<40%), cardiogenic shock, CPR on admission, peak troponin-I, NT-ProBNP, and lipid-lowering agents were common independent predictors of MACE in both the DM and non-DM groups. Appendix A shows the subgroup analysis for MACE in patients with and without DM. In the DM group, patients without cardiogenic shock, hypertension, or dyslipidemia and patients who received lipid-lowering agents or a deployed stent with a mean diameter of ≥3 mm had a lower MACE rate in the STEMI group than in the NSTEMI group. In the non-DM group, patients without dyslipidemia and those who received lipid-lowering agents had a lower MACE rate in the STEMI group than in the NSTEMI group.

## 4. Discussion

The main findings of this study were as follows: (1) In the DM group, although MACE, all-cause death, CD, recurrent MI, any repeat revascularization, and ST rates were not significantly different between the NSTEMI and STEMI groups, the non-CD rate was significantly higher in the NSTEMI group than in the STEMI group; (2) in the non-DM group, the MACE, all-cause death, and CD rates were significantly higher in the NSTEMI group than in the STEMI group; (3) in the NSTEMI group, the MACE, all-cause death, non-CD, and ST rates were significantly higher in the DM group than in the non-DM group; (4) in the STEMI group, the MACE, all-cause death, CD, recurrent MI, and any repeat revascularization rates were significantly higher in the DM group than in the non-DM group; and (5) reduced LVEF (<40%), cardiogenic shock, CPR on admission, peak troponin-I level, NT-ProBNP level, and lipid-lowering agents were common independent predictors for MACE in both the DM and non-DM groups.

STEMI is the result of acute occlusion of the IRA and is associated with transmural ischemia, whereas NSTEMI is caused by transient or incomplete coronary artery occlusion, resulting in non-transmural subendocardial ischemia [4]. In previous studies [9,31], the 6-month post-discharge mortality (6.2% vs. 4.8%, respectively) and 1-year mortality (11.6% vs. 9.0%, respectively) rates were higher in the NSTEMI group than in the STEMI group. Although these two randomized studies [9,31] are valuable for estimating comparative clinical outcomes between NSTEMI and STEMI groups, they [9,31] were conducted before the new-generation DES era and were not limited to patients with DM. Hyperglycemia contributes to increased mortality and morbidity rates through an oxidative-linked mechanism [32] and may exert significant hemodynamic effects even in normal study participants [12]. Furthermore, hyperglycemia caused by oxidative stress, inflammation, apoptosis, endothelial dysfunction, hypercoagulation, and platelet aggregation [33] could damage the ischemic myocardium in patients with NSTEMI [12]. In a study by Hao et al. [12], among 890 patients with NSTEMI who underwent PCI, hyperglycemia upon admission was an independent predictor of a 30-day (aHR, 1.014; *p* < 0.001 and aHR, 1.018, *p* < 0.001, respectively) and 3-year MACE (aHR, 1.009; *p* < 0.001 and aHR, 1.017, *p* < 0.001, respectively) in patients with and without DM. Recently, Li et al. [13] demonstrated that the incidences of all-cause death (1.1%) and MACE (3.4%) were significantly lower in patients without a history of DM and an HbA1c level of <6.5% at admission. However, DM patients with poor glycemic control at admission experienced high rates of all-cause death (18.8%) and MACE (25%) in 350 consecutive patients with STEMI during a 2-year follow-up period. Similarly, in our study, both in the NSTEMI and STEMI groups, the rates of all-cause death (aHR, 1.701; *p* = 0.002 and aHR; 1.869, *p* < 0.001, respectively) and MACE (aHR, 1.326; *p* = 0.007 and aHR, 1.481; *p* < 0.001, respectively) were significantly higher in the DM group than in the non-DM group (Table 3). However, the study population in these studies [12,13] was limited to patients with NSTEMI or STEMI. DESs have been developed to overcome the limitations of BMS deployment, such as neointimal hyperplasia and repeat revascularization [22,24]. In the era of DES, second-generation DES is the most commonly used DES because it can solve the problems of 1G-DES, such as inflammation and restenosis, and decrease the mortality rate (aHR, 1.534; *p* = 0.009) [14]. Hence, to provide more meaningful results and compensate for the shortcomings of the previous studies [9,12,13,31], we compared the 2-year clinical outcomes between the NSTEMI and STEMI groups according to the presence or absence of DM. The study population was strictly confined to patients with AMI who underwent successful implantation of a new-generation DES to reflect the current PCI trend. Additionally, to evaluate the long-term outcomes of the NSTEMI and STEMI groups, we excluded patients who died in the hospital (Figure 1).

Patients with NSTEMI tend to be older and have a lower rate of acute revascularization than those with STEMI [34]. In our study, in both the DM and non-DM groups, the patients in the NSTEMI group had a higher mean age than those in the STEMI group (65.5 ± 11.3 years vs. 63.0 ± 11.9, *p* < 0.001 and 63.5 ± 12.6 years vs. 61.4 ± 13.0, *p* < 0.001, respectively) and the total number of patients who underwent PCI within 24 h was lower in the NSTEMI group than that in the STEMI group (85.0% vs. 96.5%, *p* < 0.001 and 87.5% vs. 96.7%, *p* < 0.001, respectively, Table 1). Although this study did not include in-hospital outcomes, it could be stated that STEMI is a higher-risk disease in the acute phase [6,7], for which we have gathered treatment knowledge, while NSTEMI is a higher-risk disease in the chronic phase, owing to the higher complexity of these patients [9,10]. Moreover, in our study, although HbA1c levels were higher in patients with STEMI, the number of patients undergoing insulin treatment was higher in the NSTEMI group. Patients with NSTEMI may have had greater decompensated diabetes at the time of MI. Insulin-treated patients with DM were associated with significantly higher short- and long-term adverse cardiovascular outcomes after PCI than those not treated with insulin therapy [35]. However, long-term comparative results between patients with insulin-treated DM and non-insulin-treated DM after NSTEMI and STEMI are very limited. Further studies are required to evaluate long-term clinical outcomes in the NSTEMI and STEMI groups. Okura et al. reported that the non-CD rate (15.1% vs. 8.4 %, *p* < 0.001) was significantly higher in the NSTEMI group than in the STEMI group during a median 4.3-year follow-up period [34]. In a recent study involving patients with chronic kidney disease [36], the non-CD rate (aHR, 1.960; *p* = 0.004) was significantly higher in the NSTEMI group than in the STEMI group. In our study, after both multivariable- (aHR, 2.223; *p* = 0.006) and PS-adjusted (aHR, 2.484; *p* = 0.004) analyses, the non-CD rate was significantly higher in the DM group than that in the STEMI group (Table 2). Furthermore, after multivariable-adjusted analysis, the non-CD rate in the NSTEMI group was significantly higher in the DM group than that in the non-DM group (aHR, 2.810; *p* < 0.001, Table 3). The main causes of non-CD in the DM group were multiple organ failure (0.7% vs. 0.2%, *p* = 0.008) and CVA (0.9% vs. 0.4%, *p* = 0.034) (Appendix A). In a previous report [37], CVA was independently predicted by DM (HR, 1.43; 95% CI, 1.04-1.97; *p* = 0.03) and was associated with a significantly increased 3-year mortality rate (aHR, 2.39; *p* = 0.004). DM accelerates atherosclerosis in multiple vascular beds and causes severe coronary atherosclerosis. Coronary and cerebrovascular diseases frequently coexist because of their similar pathogeneses [38]. In our study, in the non-DM group, the mortality rate (all-cause and CD) was significantly higher in the NSTEMI group than in the STEMI group; however, in the DM group, the mortality rate did not significantly differ between the two MI groups (Table 2). This result may be related to the relatively higher CD rate in the DM and STEMI groups than that in the non-DM and STEMI groups (aHR, 2.248; *p* < 0.001, Table 3). Additionally, this result may be related to the insignificantly different non-CD rates in the DM and STEMI groups compared to those in the non-DM and STEMI groups (aHR, 1.307; *p* = 0.383, Table 3).

The incidence of recurrent ischemic events after AMI is higher in the first year, and, in subsequent years, it is based on several cardiovascular risk factors [39]. Recently, Kim et al. [40] reported that in patients with AMI, the cumulative incidence of re-MI was significantly higher in the DM group than in the normoglycemia group (aHR, 1.752; 95% CI, 1.087–2.823; *p* = 0.021). Among all of the patients in our study, the re-MI rate was significantly higher in the DM group than in the non-DM group (aHR, 1.527; 95% CI, 1.166–2.000; *p* = 0.002, Table 3). The re-MI rate increased continuously in the DM group, regardless of AMI type (Figure 2). In a Danish study [41], the risk of ST (definite, probable, or possible) did not differ significantly between DM and non-DM groups in patients with STEMI (aHR, 1.50; 95% CI, 0.92–2.45). Recently, a subgroup analysis of the ultrathin strut biodegradable polymer sirolimus-eluting stent versus the durable polymer everolimus-eluting stent for percutaneous coronary revascularization trial [42] showed that the 5-year ST (definite or probable) was significantly higher in the DM group than in the non-DM group (rate ratio (RR), 2.05; 95% CI, 1.45–2.90; *p* < 0.001) in all the patients after new-generation DES implantation. Furthermore, in their study, the target lesion failure (RR, 1.87; *p* < 0.001) and target vessel failure (rate ratio, 1.76; *p* < 0.001) rates were significantly higher in the DM group than in the non-DM group [42]. In our study, although patients in the DM with STEMI group showed comparable ST (definite or probable) rates to those in the non-DM STEMI group (aHR, 1.381; 95% CI, 0.792–2.224; *p* = 0.315; Table 3), the overall ST rate was significantly higher in the DM group than in the non-DM group (aHR, 1.521; 95% CI, 1.027–2.351; *p* = 0.037, Table 3). Additionally, the repeat revascularization rate was significantly higher in the DM group than in the non-DM group (aHR, 1.310; 95% CI, 1.069–1.605; *p* = 0.009). In our study, reduced LVEF, cardiogenic shock, CPR on admission, peak troponin-I, NT-ProBNP, and lipid-lowering agents were common independent predictors of MACE in both the DM and non-DM groups (Appendix A). These are well-known independent predictors of MACE in patients with AMI [33,43,44].

Several previous studies have demonstrated that higher long-term mortality was observed in the NSTEMI group than in the STEMI group, regardless of the type of death [7,9,10,31,45]. Consistent with the results of previous studies [7,9,10,31,45], in all of the patients in our study, the rates of MACE, all-cause death, CD, and non-CD were significantly higher in the NSTEMI group than in the STEMI group. However, the higher non-CD rate in the NSTEMI group was more evident in the DM group, while the higher CD rate in the NSTEMI group was more evident in the non-DM group. Hence, we believe that strategies for reducing the non-CD rate in patients with DM and those for reducing the CD rate in patients without DM could be beneficial for the NSTEMI group after successful PCI with new-generation DES. Although the sample size of the study population was too small to demonstrate meaningful results, more than 50 high-volume universities and community hospitals with facilities for primary PCI and on-site cardiac surgery participated in this study. Regarding the relatively higher incidence of multiple organ failure and CVA in the DM and NSTEMI groups than in the non-DM and STEMI groups (Appendix A), more well-established and regular follow-ups [9] as well as more focused and diverse secondary prevention therapies [31], including those involving lipid-lowering agents (Appendix A), are required to reduce the occurrence of non-CD in patients with NSTEMI. Since this study enrolled patients (from the registry) from 2005, many patients with diabetes did not experience the benefits of newer therapies (such as sodium–glucose cotransporter 2 inhibitor, glucagon-like peptide-1 receptor agonist). These medications have been shown to reduce the risk of cardiovascular events when compared with that of the controls [46,47]. Therefore, recent guidelines [48] support the incorporation of these newer agents with cardiovascular benefits into routine clinical practice and screening of patients who are at a high risk of cardiovascular disease. Unfortunately, because we could not obtain information about the various recently developed antidiabetic agents that had been prescribed from the KAMIR registry, we could not provide information concerning the effects of these drugs on the long-term clinical outcomes in this study. Additionally, due to the large temporal interval of this retrospective analysis, many patients did not experience the benefits brought from the newer therapies and sodium–glucose cotransporter 2 inhibitor, which was approved as a new alternative for the treatment of T2DM by the Food and Drug Administration in 2013 for use in Europe [49]. We stratified patients into two groups before and after 2013 according to the year of index myocardial infarction as shown in Table 1 and Appendix A. Finally, we believe that the results of this comparative study could provide interventional cardiologists with meaningful information regarding treatment strategies for patients with two different types of AMI according to the presence or absence of DM.

This study has several limitations. First, because we used registry data, there may have been some under-reported or missing data. Second, this study was based on discharge medications because we could not precisely determine the participants’ adherence or non-adherence to their antidiabetic drugs during the 2-year follow-up period. Third, although the interval from symptom onset to PCI was an important determinant of major clinical outcomes, this variable included many missing values in the registry data; therefore, we could not include this variable in our study. Fourth, although we performed multivariable- and PS-adjusted analyses to strengthen our results, variables not included in the KAMIR may have affected the study outcomes. Fifth, it is not certain that our population had 100% T2DM based only on the age at which diabetes was discovered. Occasionally, type 1 DM occurs in individuals aged over 30 years [50]. Moreover, there were some missing values concerning patient-reported history, such as the presence or absence of a history of ketoacidosis and other medical records indicating T2DM in the KAMIR data, owing to the registry-based nature of this study. These factors may be considered as critical limitations of this study. Sixth, this retrospective study was a long-term (November 2005 to June 2015) study of patients with AMI, which could have affected clinical outcomes. Finally, the 2-year follow-up period in this study was relatively short and may have been inadequate for determining long-term major clinical outcomes.

## 5. Conclusions

In this retrospective study, patients with NSTEMI had a significantly higher 2-year mortality rate than those with STEMI. Furthermore, strategies to reduce the non-CD rate in patients with DM and the CD rate in non-DM patients could be beneficial for those with NSTEMI. Hence, more regular follow-up and focused and diverse secondary prevention therapies to reduce the incidence of multiple organ failure and CVA are required in patients with DM and NSTEMI.

## Figures and Tables

**Figure 1 jcm-11-05079-f001:**
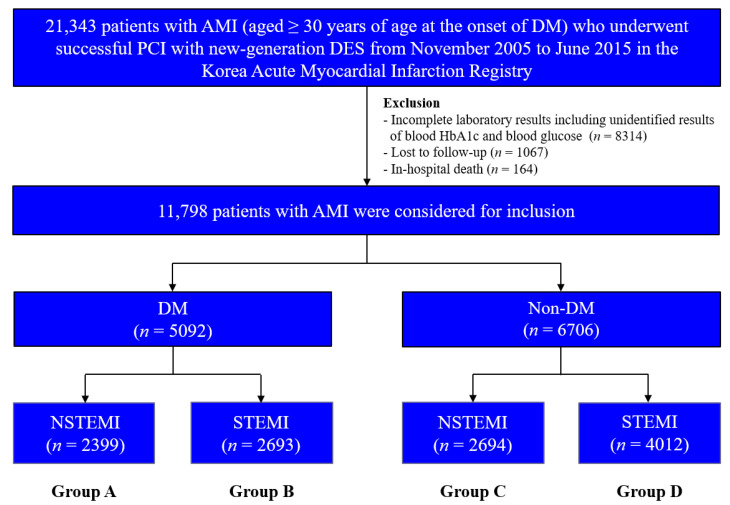
Flowchart. AMI, acute myocardial infarction; DM, diabetes mellitus; DES, drug-eluting stent; Hb, hemoglobin; NSTEMI, non-ST-segment elevation myocardial infarction; STEMI, ST-segment elevation myocardial infarction.

**Figure 2 jcm-11-05079-f002:**
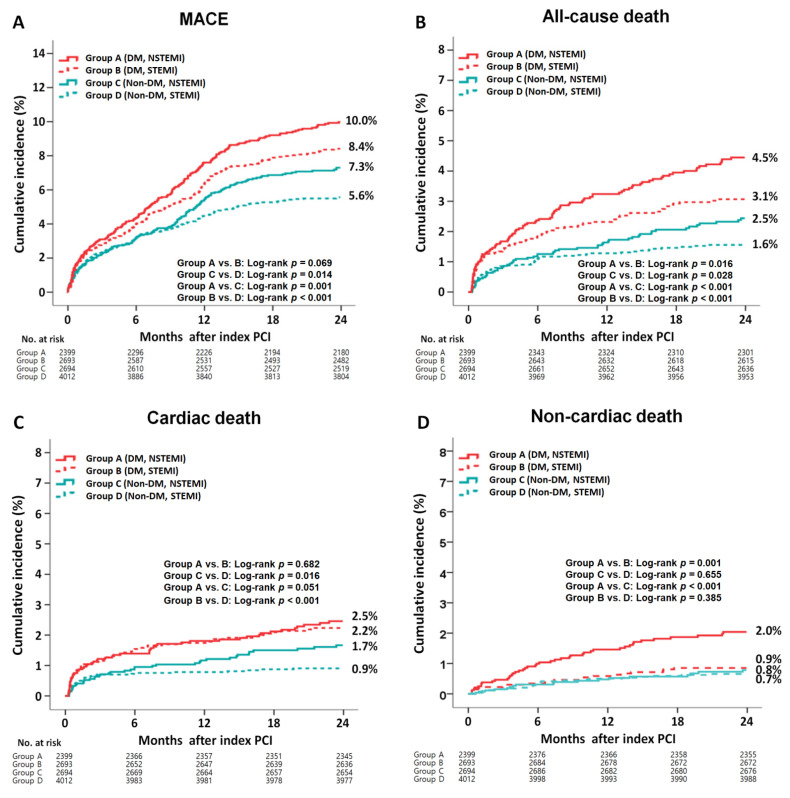
Kaplan–Meier curved analysis for MACE (**A**), all-cause death (**B**), cardiac death (**C**), non-cardiac death (**D**), recurrent MI (**E**), any repeat revascularization (**F**), and stent thrombosis (**G**). MACE, major adverse cardiac events; DM, diabetes mellitus; NSTEMI, non-ST-segment elevation myocardial infarction; STEMI, ST-segment elevation myocardial infarction; PCI, percutaneous coronary intervention.

**Table 1 jcm-11-05079-t001:** Baseline clinical, laboratory, angiographic, and procedural characteristics.

Variables	Overall (*n* = 11,798)			DM (*n* = 5092)			Non-DM (*n* = 6706)		
	NSTEMI (*n* = 5093)	STEMI (*n* = 6705)	*p* Value	NSTEMI (*n* = 2399)	STEMI (*n* = 2693)	*p* Value	NSTEMI (*n* = 2694)	STEMI (*n* = 4012)	*p* Value
Male, *n* (%)	3626 (71.2)	5184 (77.3)	<0.001	1585 (66.1)	1996 (74.1)	<0.001	2041 (75.8)	3188 (79.5)	0.001
Age, years	64.4 ± 12.0	62.1 ± 12.5	<0.001	65.5 ± 11.3	63.0 ± 11.9	<0.001	63.5 ± 12.6	61.4 ± 13.0	<0.001
LVEF, %	54.1 ± 11.1	50.7 ± 11.0	<0.001	52.6 ± 11.9	50.0 ± 11.1	<0.001	55.4 ± 10.2	51.2 ± 10.8	<0.001
BMI, kg/m^2^	24.1 ± 3.2	24.2 ± 3.1	0.076	24.3 ± 3.1	24.5 ± 3.1	0.017	24.0 ± 3.2	24.0 ± 3.1	0.404
SBP, mmHg	134.8 ± 26.6	127.7 ± 27.9	<0.001	134.5 ± 26.8	128.1 ± 28.6	<0.001	135.0 ± 26.5	127.5 ± 27.5	<0.001
DBP, mmHg	80.7 ± 15.2	78.4 ± 16.8	<0.001	79.8 ± 15.1	77.8 ± 17.0	<0.001	81.5 ± 15.7	78.7 ± 16.7	<0.001
Cardiogenic shock, *n* (%)	113 (2.2)	395 (5.9)	<0.010	61 (2.5)	176 (6.5)	<0.001	52 (1.9)	219 (5.5)	<0.001
Killip class I/II, *n* (%)	4440 (87.2)	5613 (83.7)	<0.001	2013 (83.9)	2211 (82.1)	0.087	2427 (90.1)	3402 (84.8)	<0.001
CPR on admission, *n* (%)	122 (2.4)	374 (5.6)	<0.001	60 (2.5)	128 (4.8)	<0.001	62 (2.3)	246 (6.1)	<0.001
Hypertension, *n* (%)	2779 (54.6)	3120 (46.5)	<0.001	1540 (64.2)	1532 (56.9)	<0.001	1239 (46.0)	1588 (39.6)	<0.001
Dyslipidemia, *n* (%)	661 (13.0)	736 (11.0)	0.001	375 (15.6)	356 (13.2)	0.014	286 (10.6)	380 (9.5)	0.134
Previous MI, *n* (%)	234 (4.6)	196 (2.9)	<0.001	149 (6.2)	98 (3.6)	<0.001	85 (3.2)	98 (2.4)	0.079
Previous PCI, *n* (%)	386 (7.6)	297 (4.4)	<0.001	239 (10.0)	154 (5.7)	<0.001	147 (5.5)	143 (3.6)	<0.001
Previous CABG, *n* (%)	31 (0.6)	20 (0.3)	0.015	25 (1.0)	14 (0.5)	0.036	6 (0.2)	6 (0.1)	0.561
Previous HF, *n* (%)	72 (1.4)	55 (0.8)	0.002	50 (2.1)	28 (1.0)	0.003	22 (0.8)	27 (0.7)	0.498
Previous CVA, *n* (%)	387 (7.6)	326 (4.9)	<0.001	224 (9.3)	168 (6.2)	<0.001	163 (6.1)	158 (3.9)	<0.001
Current smokers, *n* (%)	1922 (37.7)	3194 (47.6)	<0.001	803 (33.5)	1182 (43.9)	<0.001	1119 (41.5)	2012 (50.1)	<0.001
Peak CK-MB, mg/dL	23 (7.0–82.9)	124.0 (34.6–268.5)	<0.001	18.3 (6.0–63.4)	102.4 (26.9–236.2)	<0.001	29.5 (8.7–103.1)	140.5 (39.6–290.8)	<0.001
Peak Troponin-I, ng/mL	11.0 (2.2–47.8)	46.8 (16.0–61.2)	<0.001	8.7 (1.7–42.0)	48.1 (18.5–67.8)	<0.001	12.7 (2.6–47.8)	47.8 (12.7–52.1)	<0.001
Blood glucose, mg/dL	171.2 ± 88.6	184.6 ± 83.3	<0.001	216.5 ± 103.1	234.3 ± 95.9	<0.001	130.6 ± 43.4	150.8 ± 50.8	<0.001
Hemoglobin A1c, %	6.64 ± 2.08	6.57 ± 2.22	0.074	7.72 ± 2.60	7.90 ± 3.00	0.022	5.67 ± 0.45	5.67 ± 0.45	0.800
NT-ProBNP, pg/mL	658.0 (160.0–2740.5)	265.0 (59.0–1443.0)	<0.001	1325.5 (301.0–5417.0)	453.5 (89.0–2621.8)	<0.001	484.0 (130.5–1857.5)	225.0 (53.0–1195.0)	<0.001
Hs-CRP, mg/dL	10.1 ± 46.8	9.4 ± 37.9	0.427	10.7 ± 43.9	11.3 ± 43.1	0.596	9.5 ± 49.2	8.2 ± 33.9	0.210
Serum creatinine, mg/dL	1.17 ± 1.70	1.06 ± 1.20	<0.001	1.34 ± 2.21	1.10 ± 0.86	<0.001	1.03 ± 1.06	1.03 ± 1.39	0.822
eGFR, mL/min/1.73 m^2^	88.3 ± 45.3	87.3 ± 37.2	0.215	83.1 ± 46.0	85.3 ± 41.3	0.074	92.8 ± 44.1	88.6 ± 34.1	<0.001
Total cholesterol, mg/dL	181.8 ± 46.3	184.0 ± 43.7	0.013	177.3 ± 49.8	180.5 ± 45.8	0.015	186.0 ± 42.4	186.3 ± 42.1	0.795
Triglyceride, mg/L	136.3 ± 117.5	137.9 ± 109.8	0.445	150.2 ± 136.2	151.0 ± 124.0	0.814	123.9 ± 96.3	129.0 ± 98.2	0.032
HDL cholesterol, mg/L	43.0 ± 14.1	43.3 ± 15.4	0.291	41.6 ± 13.5	42.0 ± 14.4	0.359	44.2 ± 14.4	44.1 ± 16.0	0.869
LDL cholesterol, mg/L	114.7 ± 41.7	115.9 ± 39.5	0.096	109.3 ± 38.8	111.8 ± 37.2	0.018	119.5 ± 43.6	118.7 ± 40.8	0.467
Diabetes management									
Diet, *n* (%)	166 (3.3)	244 (3.6)	0.287	166 (6.9)	244 (9.0)	0.005			
Oral agent, *n* (%)	1492 (29.3)	1681 (25.1)	<0.001	1492 (62.2)	1681 (62.4)	0.885			
Insulin, *n* (%)	160 (3.1)	128 (1.9)	<0.001	160 (6.7)	128 (4.8)	0.003			
Untreated, *n* (%)	581 (11.4)	640 (9.5)	0.001	581 (24.2)	640 (23.8)	0.718			
Discharge medications									
Aspirin, *n* (%)	4940 (97.2)	6484 (96.7)	0.370	2326 (97.0)	2596 (96.4)	0.268	2614 (97.0)	3888 (96.9)	0.777
Clopidogrel, *n* (%)	4325 (84.9)	5780 (86.2)	0.049	2115 (88.2)	2335 (86.7)	0.118	2210 (82.0)	3445 (85.9)	<0.001
Ticagrelor, *n* (%)	484 (9.5)	607 (9.1)	0.403	183 (7.6)	219 (8.1)	0.532	301 (11.2)	388 (9.7)	0.047
Prasugrel, *n* (%)	236 (4.6)	366 (5.5)	0.047	101 (4.2)	139 (5.2)	0.112	135 (4.4)	227 (5.7)	0.270
Cilostazol, *n* (%)	887 (17.4)	1242 (18.5)	0.122	462 (19.3)	538 (20.0)	0.525	425 (15.8)	704 (17.5)	0.058
ACEIs, *n* (%)	2581 (50.7)	3849 (57.4)	<0.001	1133 (47.2)	1455 (54.0)	<0.001	1448 (53.7)	2394 (59.7)	<0.001
ARBs, *n* (%)	1558 (30.6)	1576 (23.5)	<0.001	828 (34.5)	712 (26.4)	<0.001	730 (27.1)	864 (21.5)	<0.001
BBs, *n* (%)	4219 (82.8)	5641 (84.1)	0.061	2007 (83.7)	2266 (84.1)	0.639	2212 (82.1)	3375 (84.1)	0.030
CCBs, *n* (%)	501 (9.8)	243 (3.6)	<0.001	265 (11.0)	115 (4.3)	<0.001	236 (8.8)	128 (3.2)	<0.001
Lipid lowering agents, *n* (%)	4427 (86.9)	5746 (85.7)	0.056	2038 (85.0)	2262 (84.0)	0.347	2389 (88.7)	3484 (86.8)	0.025
Year of index MI			<0.001			<0.001			<0.001
Before 2013	2959 (58.1)	4372 (65.2)		1480 (61.7)	1834 (68.1)		1479 (54.9)	2538 (63.3)	
After 2013	2134 (41.9)	2333 (34.8)		919 (38.3)	859 (31.9)		1215 (45.1)	1474 (36.7)	
IRA									
Left main, *n* (%)	127 (2.5)	75 (1.1)	<0.001	66 (2.8)	31(1.2)	<0.001	61 (2.3)	44 (1.1)	<0.001
LAD, *n* (%)	2178 (42.8)	3548 (52.9)	<0.001	1006 (41.9)	1343 (49.9)	<0.001	1172 (43.5)	2205 (55.0)	<0.001
LCx, *n* (%)	1364 (26.8)	592 (8.8)	<0.001	625 (26.1)	233 (8.7)	<0.001	739 (27.4)	359 (8.9)	<0.001
RCA, *n* (%)	1424 (28.0)	2490 (37.1)	<0.001	702 (29.3)	1086 (40.3)	<0.001	722 (26.8)	1404 (35.0)	<0.001
Treated vessel									
Left main, *n* (%)	214 (4.2)	115 (1.7)	<0.001	105 (4.4)	44 (1.6)	<0.001	109 (4.0)	71 (1.8)	<0.001
LAD, *n* (%)	2888 (56.7)	4035 (60.2)	<0.001	1378 (57.4)	1582 (58.7)	0.346	1510 (56.1)	2453 (61.1)	<0.001
LCx, *n* (%)	2016 (39.6)	1076 (16.0)	<0.001	957 (39.9)	454 (16.9)	<0.001	1059 (39.3)	622 (15.5)	<0.001
RCA, *n* (%)	1868 (36.7)	2824 (42.1)	<0.001	937 (39.1)	1236 (45.9)	<0.001	931 (34.6)	1588 (39.6)	<0.001
ACC/AHA lesion type									
Type B1, *n* (%)	703 (13.8)	857 (12.8)	0.105	327 (13.6)	327 (12.1)	0.113	376 (14.0)	530 (13.2)	0.381
Type B2, *n* (%)	1797 (35.3)	2031 (30.3)	<0.001	835 (34.8)	825 (30.6)	0.002	962 (35.7)	1206 (30.1)	<0.001
Type C, *n* (%)	2124 (41.7)	3139 (46.8)	<0.001	1025 (42.7)	1280 (47.5)	0.001	1099 (40.8)	1859 (46.3)	<0.001
Extent of CAD									
1-vessel, *n* (%)	2208 (43.4)	3551 (53.0)	<0.001	910 (37.9)	1267 (47.0)	<0.001	1298 (48.2)	2284 (56.9)	<0.001
2-vessel, *n* (%)	1724 (33.9)	2010 (30.0)	<0.001	848 (35.3)	845 (31.4)	0.003	876 (32.6)	1165 (29.0)	0.002
≥3-vessel, *n* (%)	1161 (22.8)	1144 (17.1)	<0.001	641 (26.7)	581 (21.6)	<0.001	520 (19.3)	563 (14.0)	<0.001
Pre-PCI TIMI 0/1, *n* (%)	2047 (40.2)	4781 (71.3)	<0.001	903 (37.6)	1884 (70.0)	<0.001	1144 (42.5)	2897 (72.2)	<0.001
PCI within 24 h, *n* (%)	4396 (86.3)	6477 (96.6)	<0.001	2040 (85.0)	2599 (96.5)	<0.001	2356 (87.5)	3878 (96.7)	<0.001
GP IIb/IIIa inhibitor, n (%)	480 (9.4)	1458 (21.7)	<0.001	202 (8.4)	527 (19.6)	<0.001	278 (10.3)	931 (23.2)	<0.001
Transradial approach, *n* (%)	1833 (36.0)	1218 (18.2)	<0.001	815 (34.0)	493 (18.3)	<0.001	1018 (37.8)	725 (18.1)	<0.001
IVUS, *n* (%)	1241 (24.4)	1352 (20.2)	<0.001	539 (22.5)	542 (20.1)	0.041	702 (26.1)	810 (20.2)	<0.001
OCT, *n* (%)	67 (1.3)	22 (0.3)	<0.001	26 (1.1)	10 (0.4)	0.004	41 (1.5)	12 (0.3)	<0.001
FFR, *n* (%)	75 (1.5)	61 (0.9)	0.448	38 (1.6)	24 (0.9)	0.029	37 (1.4)	37 (0.9)	0.095
Types of DES ^a^									
ZES, *n* (%)	1647 (32.3)	2372 (35.4)	0.206	789 (32.9)	976 (36.2)	0.012	858 (31.8)	1396 (34.8)	0.012
EES, *n* (%)	2710 (53.2)	3374 (50.3)	0.193	1289 (53.7)	1336 (49.6)	0.003	1421 (52.7)	2038 (50.8)	0.117
BES, *n* (%)	783 (15.4)	873 (13.0)	0.078	335 (14.0)	330 (12.3)	0.071	448 (16.6)	543 (13.5)	<0.001
Others, *n* (%)	124 (2.4)	190 (2.8)	0.710	72 (3.0)	89 (3.3)	0.575	52 (1.9)	101 (2.5)	0.133
Stent diameter, mm	3.18 ± 0.42	3.07 ± 0.42	<0.001	3.15 ± 0.42	3.04 ± 0.41	<0.001	3.19 ± 0.41	3.09 ± 0.42	<0.001
Stent length, mm	27.8 ± 12.8	26.7 ± 10.2	0.019	28.0 ± 12.9	27.0 ± 10.5	0.002	27.6 ± 12.6	26.5 ± 10.0	<0.001
Number of stents	1.61 ± 0.89	1.40 ± 0.70	<0.001	1.67 ± 0.90	1.44 ± 0.74	<0.001	1.56 ± 0.88	1.40 ± 0.66	<0.001

Values are means ± standard deviation or median (interquartile range) or numbers and percentages. The *p* values for continuous data were obtained from the unpaired *t*-test. The *p* values for categorical data were obtained from the chi-squared or Fisher’s exact test. DM, diabetes mellitus; NSTEMI, non-ST-segment elevation myocardial infarction; STEMI, ST-segment elevation myocardial infarction; LVEF, left ventricular ejection fraction; BMI, body mass index; SBP, systolic blood pressure; DBP, diastolic blood pressure; CPR, cardiopulmonary resuscitation; PCI, percutaneous coronary intervention; CABG, coronary artery bypass graft; HF, heart failure; CVA, cerebrovascular accidents; eGFR, estimated glomerular filtration rate; CK-MB, creatine kinase myocardial band; NT-ProBNP, N-terminal pro-brain natriuretic peptide; Hs-CRP, high sensitivity-C-reactive protein; HDL, high-density lipoprotein; LDL, low-density lipoprotein; ACEIs, angiotensin converting enzyme inhibitors; ARBs, angiotensin receptor blockers; BBs, beta-blockers; CCBs, calcium channel blockers; LAD, left anterior descending coronary artery; LCx, left circumflex coronary artery; RCA, right coronary artery; ACC/AHA, American College of Cardiology/American Heart Association; CAD, coronary artery disease; TIMI, thrombolysis in myocardial infarction; ZES, zotarolimus-eluting stent; EES, everolimus-eluting stent; BES, biolimus-eluting stent; GP, glycoprotein; IVUS, intravascular ultrasound; OCT, optical coherence tomography; FFR, fractional flow reserve; ^a^ Drug-eluting stents were composed of ZES (Resolute Integrity stent; Medtronic, Inc., Minneapolis, MN, USA), EES (Xience Prime stent, Abbott Vascular, Santa Clara, CA, USA; or Promus Element stent, Boston Scientific, Natick, MA, USA), and BES (BioMatrix Flex stent, Biosensors International, Morges, Switzerland; or Nobori stent, Terumo Corporation, Tokyo, Japan).

**Table 2 jcm-11-05079-t002:** Clinical outcomes between NSTEMI and STEMI groups at 2 years.

Outcomes	DM (*n* = 5092)		Log-Rank	Unadjusted		Multivariable-Adjusted ^a^		Propensity Score-Adjusted	
	NSTEMI (*n* = 2399)	STEMI (*n* = 2693)		HR (95% CI)	*p*	HR (95% CI)	*p*	HR (95% CI)	*p*
MACE	219 (10.0)	211 (8.4)	0.069	1.192 (0.986–1.440)	0.069	1.098 (0.875–1.314)	0.401	1.074 (0.850–1.358)	0.548
All-cause death	98 (4.5)	78 (3.1)	0.016	1.436 (1.067–1.934)	0.017	1.275 (0.914–1.684)	0.121	1.334 (0.926–1.922)	0.102
Cardiac death	54 (2.5)	57 (2.2)	0.682	1.081 (0.745–1.568)	0.682	1.068 (0.695–1.580)	0.756	1.078 (0.720–1.690)	0.732
Non-cardiac death	44 (2.0)	21 (0.9)	0.001	2.403 (1.429–4.041)	0.001	2.200 (1.231–3.813)	0.007	2.484 (1.326–4.651)	0.004
Recurrent MI	61 (2.9)	59 (2.4)	0.339	1.191 (0.832–1.703)	0.339	1.102 (0.764–1.612)	0.580	1.104 (0.751–1.720)	0.660
Any repeat revascularization	95 (4.5)	99 (4.1)	0.484	1.106 (0.835–1.465)	0.484	1.021 (0.612–1.312)	0.901	1.186 (0.839–1.675)	0.334
ST (definite or probable)	20 (0.8)	25 (0.9)	0.719	0.898 (0.499–1.616)	0.719	0.901 (0.512–1.702)	0.745	0.946 (0.485–1.968)	0.882
**Outcomes**	**Non-DM (*n* = 6706)**		**Log-Rank**	**Unadjusted**		**Multivariable-Adjusted ^a^**		**Propensity score-Adjusted**	
	**NSTEMI** **(*n* = 2694)**	**STEMI** **(*n* = 4012)**		**HR (95% CI)**	** *p* **	**HR (95% CI)**	** *p* **	**HR (95% CI)**	** *p* **
MACE	175 (7.3)	208 (5.6)	0.014	1.287 (1.052–1.574)	0.014	1.384 (1.120–1.694)	0.002	1.543 (1.211–1.965)	<0.001
All-cause death	58 (2.5)	59 (1.6)	0.028	1.497 (1.042–2.151)	0.029	2.054 (1.399–3.031)	<0.001	2.172 (1.423–3.276)	<0.001
Cardiac death	40 (1.7)	35 (0.9)	0.016	1.734 (1.102–2.730)	0.017	2.688 (1.698–4.331)	<0.001	2.882 (1.679–4.967)	<0.001
Non-cardiac death	18 (0.8)	24 (0.7)	0.655	1.150 (0.624–2.118)	0.655	1.215 (0.575–2.182)	0.598	1.250 (0.641–2.557)	0.539
Recurrent MI	46 (1.9)	57 (1.5)	0.300	1.227 (0.832–1.810)	0.301	1.194 (0.772–1.751)	0.511	1.380 (0.865–2.201)	0.177
Any repeat revascularization	84 (3.6)	108 (3.0)	0.225	1.193 (0.897–1.586)	0.226	1.184 (0.788–1.532)	0.278	1.299 (0.920–1.835)	0.137
ST (definite or probable)	9 (0.3)	27 (0.7)	0.063	0.495 (0.233–1.053)	0.068	0.623 (0.302–1.178)	0.184	0.513 (0.252–1.245)	0.140
**Outcomes**	**Overall (*n* = 11,798)**		**Log-Rank**	**Unadjusted**		**Multivariable-Adjusted ^a^**		**Propensity score-Adjusted**	
	**NSTEMI** **(*n* = 5093)**	**STEMI** **(*n* = 6705)**		**HR (95% CI)**	** *p* **	**HR (95% CI)**	** *p* **	**HR (95% CI)**	** *p* **
MACE	394 (8.6)	419 (6.7)	0.001	1.269 (1.106–1.456)	0.001	1.214 (1.034–1.315)	0.006	1.298 (1.097–1.535)	0.002
All-cause death	156 (3.4)	137 (2.2)	<0.001	1.529 (1.216–1.924)	<0.001	1.521 (1.208–1.994)	<0.001	1.653 (1.252–2.183)	<0.001
Cardiac death	94 (2.0)	92 (1.4)	0.032	1.369 (1.027–1.824)	0.032	1.367 (1.009–1.684)	0.041	1.499 (1.060–2.120)	0.022
Non-cardiac death	62 (1.4)	45 (0.7)	0.001	1.859 (1.266–2.729)	0.002	1.745 (1.207–2.596)	0.005	1.977 (1.239–3.155)	0.004
Recurrent MI	107 (2.4)	116 (1.9)	0.102	1.245 (0.957–1.619)	0.103	1.214 (0.902–1.563)	0.372	1.247 (0.905–1.719)	0.177
Any repeat revascularization	179 (4.0)	207 (3.4)	0.123	1.170 (0.958–1.429)	0.123	1.060 (0.682–1.297)	0.521	1.068 (0.837–1.324)	0.597
ST (definite or probable)	29 (0.6)	52 (0.8)	0.179	0.733 (0.466–1.155)	0.181	0.746 (0.502–1.273)	0.382	0.754 (0.433–1.313)	0.319

MACE, major adverse cardiac events; DM, diabetes mellitus; NSTEMI, non-ST-segment elevation myocardial infarction; STEMI, ST-segment elevation myocardial infarction; HR, hazard ratio; CI, confidence interval; LVEF, left ventricular ejection fraction; BMI, body mass index; SBP, systolic blood pressure; DBP, diastolic blood pressure; CPR, cardiopulmonary resuscitation; PCI, percutaneous coronary intervention; CABG, coronary artery bypass graft; HF, heart failure; CVA, cerebrovascular accidents; CK-MB, creatine kinase myocardial band; NT-ProBNP, N-terminal pro-brain natriuretic peptide; eGFR, estimated glomerular filtration rate; LDL, low-density lipoprotein; ACEI, angiotensin-converting enzyme inhibitor; ARB, angiotensin receptor blocker; BB, beta-blocker; CCB, calcium channel blocker. ^a^ Adjusted by male sex, age, LVEF, BMI, SBP, DBP, cardiogenic shock, Killip class I/II, CPR on admission, hypertension, dyslipidemia, previous histories (MI, PCI, CABG, HF, and CVA), current smoker, peak CK-MB, peak troponin-I, NT-ProBNP, serum creatinine, eGFR, total cholesterol, triglyceride, LDL cholesterol, clopidogrel, ticagrelor, prasugrel, ACEI, ARB, BB, CCB, lipid-lowering agents, and year of index MI (Appendix A).

**Table 3 jcm-11-05079-t003:** Clinical outcomes between the DM and non-DM groups at 2 years.

Outcomes	NSTEMI (*n* = 5093)		Log-Rank	Unadjusted		Multivariable-Adjusted ^a^	
	DM (*n* = 2399)	Non-DM (*n* = 2694)		HR (95% CI)	*p*	HR (95% CI)	*p*
MACE	219 (10.0)	175 (7.3)	0.001	1.391 (1.140–1.696)	0.001	1.326 (1.080–1.629)	0.007
All-cause death	98 (4.5)	58 (2.5)	<0.001	1.873 (1.354–2.592)	<0.001	1.701 (1.215–2.382)	0.002
Cardiac death	54 (2.5)	40 (1.7)	0.051	1.498 (0.995–2.255)	0.053	1.325 (0.866–2.027)	0.195
Non-cardiac death	44 (2.0)	18 (0.8)	<0.001	2.706 (1.564–4.683)	<0.001	2.549 (1.450–4.480)	0.001
Recurrent MI	61 (2.9)	46 (1.9)	0.048	1.469 (1.002–2.154)	0.049	1.469 (0.990–2.180)	0.056
Any repeat revascularization	95 (4.5)	84 (3.6)	0.124	1.259 (0.938–1.688)	0.125	1.223 (0.904–1.654)	0.191
ST (definite or probable)	20 (0.8)	9 (0.3)	0.018	2.501 (1.139–5.492)	0.022	2.272 (1.010–5.023)	0.048
**Outcomes**	**STEMI (*n* = 6705)**		**Log-rank**	**Unadjusted**		**Multivariable-Adjusted ^a^**	
	**DM** **(*n* = 2693)**	**Non-DM** **(*n* = 4012)**		**HR (95% CI)**	** *p* **	**HR (95% CI)**	** *p* **
MACE	211 (8.4)	208 (5.6)	<0.001	1.507 (1.244–1.825)	<0.001	1.481 (1.218–1.801)	<0.001
All-cause death	78 (3.1)	59 (1.6)	<0.001	1.961 (1.399–2.751)	<0.001	1.869 (1.322–2.643)	<0.001
Cardiac death	57 (2.2)	35 (0.9)	<0.001	2.419 (1.588–3.684)	<0.001	2.248 (1.462–3.458)	<0.001
Non-cardiac death	21 (0.9)	24 (0.7)	0.385	1.296 (0.721–2.327)	0.386	1.307 (0.716–2.384)	0.383
Recurrent MI	59 (2.4)	57 (1.5)	0.020	1.535 (1.067–2.209)	0.021	1.537 (1.060–2.228)	0.023
Any repeat revascularization	99 (4.1)	108 (3.0)	0.027	1.360 (1.035–1.786)	0.027	1.374 (1.041–1.816)	0.024
ST (definite or probable)	25 (0.9)	27 (0.7)	0.244	1.380 (0.801–2.377)	0.246	1.381 (0.792–2.224)	0.315
**Outcomes**	**Overall (*n* = 11,798)**		**Log-rank**	**Unadjusted**		**Multivariable-Adjusted ^a^**	
	**DM** **(*n* = 5092)**	**Non-DM** **(*n* = 6706)**		**HR (95% CI)**	** *p* **	**HR (95% CI)**	** *p* **
MACE	430 (9.1)	383 (6.2)	<0.001	1.473 (1.283–1.690)	<0.001	1.434 (1.245–1.652)	<0.001
All-cause death	176 (3.7)	117 (1.9)	<0.001	1.969 (1.558–2.488)	<0.001	1.866 (1.467–2.374)	<0.001
Cardiac death	111 (2.3)	75 (1.2)	<0.001	1.939 (1.447–2.599)	<0.001	1.818 (1.345–2.457)	<0.001
Non-cardiac death	65 (1.4)	42 (0.7)	<0.001	2.022 (1.372–2.981)	<0.001	1.957 (1.313–2.916)	0.001
Recurrent MI	120 (2.7)	103 (1.7)	0.002	1.524 (1.171–1.983)	0.002	1.527 (1.166–2.000)	0.002
Any repeat revascularization	194 (4.0)	192 (3.4)	0.006	1.324 (1.085–1.617)	0.006	1.314 (1.071–1.611)	0.009
ST (definite or probable)	45 (0.9)	36 (0.5)	0.024	1.648 (1.063–2.554)	0.026	1.521 (1.027–2.351)	0.037

NSTEMI, non-ST-segment elevation myocardial infarction; STEMI, ST-segment elevation myocardial infarction; DM, diabetes mellitus; HR, hazard ratio; CI, confidence interval; MACE, major adverse cardiac events; ST, stent thrombosis; LVEF, left ventricular ejection fraction; BMI, body mass index; SBP, systolic blood pressure; DBP, diastolic blood pressure; CPR, cardiopulmonary resuscitation; PCI, percutaneous coronary intervention; CABG, coronary artery bypass graft; HF, heart failure; CVA, cerebrovascular accidents; CK-MB, creatine kinase myocardial band; NT-ProBNP, N-terminal pro-brain natriuretic peptide; eGFR, estimated glomerular filtration rate; LDL, low-density lipoprotein; ACEI, angiotensin converting enzyme inhibitor. ^a^ Adjusted by male sex, age, LVEF, BMI, SBP, DBP, cardiogenic shock, Killip class I/II, CPR on admission, hypertension, previous histories (MI, PCI, CABG, HF, and CVA), current smoker, peak CK-MB, peak troponin-I, NT-ProBNP, serum creatinine, eGFR, total cholesterol, LDL cholesterol, and year of index MI (Appendix A).

## Data Availability

Data is contained within the article or Appendix A.

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
