# Peer review of "Comparison of Clinical Outcomes after Non-ST-Segment and ST-Segment Elevation Myocardial Infarction in Diabetic and Nondiabetic Populations"

_jcm, 2022, doi:10.3390/jcm11175079_

Round 1
Reviewer 1 Report
The study from Yong Hoon Kim et al is very interesting and well conducted. Although, it is not the first study concluding for a worse outcome in patients presenting with NSTEMI than those with STEMI, it has, in my opinion, an interest because interventional treatment with bare metal stents was excluded. Despite of the observational nature of the study, authors analyzed more the 11 thousands of patients with a well conducted statistical analysis.
- From the survival analysis provided it seems that the difference in long term MACE was driven only by an higher incidence of non cardiac death in NSTEMI diabetic patients. The effect on non cardiac mortality of diabetes doesn’t seem to affect STEMI patients. A quick interpretation could be based on age and comorbilities that affect more NSTEMI patients than STEMI ones. Although this study didn’t include in-hospital outcomes, from my point of view STEMI is a more dangerous disease in the acute phase, that we have learnt how to treat, while NSTEMI is a more dangerous disease in the chronic phase, due to a higher complexity of these patients. What do authors think about this point?
- Authors provided values of glycosilated hemoglobin. HA1c (not at a target level in both groups) was higher in STEMI patients, while insulin treatment was higher in NSTEMI group. Is it possible that NSTEMI patients had, at the time of the MI, a more decompensated diabetes? If yes, could have influenced the non CV death among the 2 groups?
- Since this study includes patients enrolled from 2005 in the registry, many diabetic patients didn’t experimented the benefits brought from the newer therapies (SGLT2i, GLP1a, etc). This point is well expressed in the limitations, but I think could be a good point to cite in the discussion section where authors talk about strategies to reduce non CV death in NSTEMI and DM patients.
- From statistical methods it seems that a test for normal distribution wasn’t performed, indeed some variables had a SD higher than the mean value (I.e. NTproBNP). This results let me think for a non normal distribution of the variable. Please fix the issue, reporting median and interquartile range.
- Authors provided several KM curves. KM curves from A to D show a cumulative incidence scale ranging from 0 to 8, while E-G curves from 0 to 6. Please uniform the scales.
Author Response
Comments and Suggestions for Authors
The study from Yong Hoon Kim et al is very interesting and well conducted. Although, it is not the first study concluding for a worse outcome in patients presenting with NSTEMI than those with STEMI, it has, in my opinion, an interest because interventional treatment with bare metal stents was excluded. Despite of the observational nature of the study, authors analyzed more the 11 thousands of patients with a well conducted statistical analysis.
Answer: We sincerely thank you for reviewer’s valuable comments
- From the survival analysis provided it seems that the difference in long term MACE was driven only by an higher incidence of non cardiac death in NSTEMI diabetic patients. The effect on non cardiac mortality of diabetes doesn’t seem to affect STEMI patients. A quick interpretation could be based on age and comorbilities that affect more NSTEMI patients than STEMI ones. Although this study didn’t include in-hospital outcomes, from my point of view STEMI is a more dangerous disease in the acute phase, that we have learnt how to treat, while NSTEMI is a more dangerous disease in the chronic phase, due to a higher complexity of these patients. What do authors think about this point?
Answer: We sincerely thank you for reviewer’s valuable comments and we totally agree with the reviewer’s suggestion. We cited above reviewer’s comments in the “Discussion” section as follows,
Before,
- Discussion
The main findings of this study were as follows: (1) in the DM group, although MACE, all-cause death, CD, recurrent MI, any repeat revascularization, and ST rates were not significantly different between the NSTEMI and STEMI groups, the non-CD rate was significantly higher in the NSTEMI group than in the STEMI group; (2) in the non-DM group, the MACE, all-cause death, and CD rates were significantly higher in the NSTEMI group than in the STEMI group; (3) in the NSTEMI group, the MACE, all-cause death, non-CD, and ST rates were significantly higher in the DM group than in the non-DM group; (4) in the STEMI group, the MACE, all-cause death, CD, recurrent MI, and any repeat revascularization rates were significantly higher in the DM group than in the non-DM group; and (5) reduced LVEF (<40%), cardiogenic shock, CPR on admission, peak troponin-I level, NT-ProBNP level, and lipid-lowering agents were common independent predictors for MACE in both the DM and non-DM groups.
STEMI is the result of acute occlusion of the IRA and is associated with transmural ischemia, whereas NSTEMI is caused by transient or incomplete coronary artery occlusion, resulting in non-transmural subendocardial ischemia [4]. In the previous studies [9,31], the 6-month after post-discharge mortality (6.2% vs. 4.8%, respectively) and 1-year mortality (11.6% vs. 9.0%, respectively) were higher in the NSTEMI group than that in the STEMI group. Although these two randomized studies [9,31] are valuable for estimating comparative clinical outcomes between the NSTEMI and STEMI groups, they [9,31] were conducted before the new-generation DES era and were not limited to patients with DM. Hyperglycemia contributes to increased mortality and morbidity rates through an oxidative-linked mechanism [32] and may exert significant hemodynamic effects even in normal subjects [12]. Furthermore, hyperglycemia caused by oxidative stress, inflammation, apoptosis, endothelial dysfunction, hypercoagulation, and platelet aggregation [33] could damage the ischemic myocardium in patients with NSTEMI [12]. In a study by Hao et al. [12], among 890 NSTEMI patients who underwent PCI, admission hyperglycemia was an independent predictor of 30-day (aHR, 1.014; p <0.001 and aHR, 1.018, p <0.001, respectively) and 3-year MACE (aHR, 1.009; p <0.001 and aHR, 1.017, p <0.001, respectively) in patients with and without DM. Recently, Li et al. [13] demonstrated that the incidence of all-cause death (1.1%) and MACE (3.4%) was significantly the low in patients without a DM history and admission HbA1c level < 6.5%. However, DM patients with poor glycemic control on admission experienced the high rates of all-cause death (18.8%) and MACE (25%) in their 350 consecutive patients with STEMI during a 2-year follow-up period. Similarly, in our study, both in the NSTEMI group and in the STEMI group, the rates of all-cause death (aHR, 1.755; p = 0.001 and aHR; 2.040, p <0.001, respectively) and MACE (aHR, 1.328; p = 0.006 and aHR, 1.527; p <0.001, respectively) were significantly higher in the DM group than in the non-DM group (Table 3). However, the study population in these studies [12,13] was limited to patients with NSTEMI or STEMI, separately. DESs were developed to overcome the limitations of BMS deployment, such as neointimal hyperplasia and repeat revascularization [22,24]. In the era of DES, second-generation DES (2G-DES) is the most commonly used DES because it can solve the problems of 1G-DES, such as inflammation and restenosis, and can decrease the mortality rate (aHR, 1.534; p = 0.009) [14]. Hence, to provide more meaningful results and compensate for the shortcomings of the previous studies [9,12,13,31] from the current study, we compared the 2-year clinical outcomes between the NSTEMI and STEMI groups according to the presence or absence of DM. The study population was strictly confined to patients with AMI who underwent successful implantation of new-generation DES to reflect the current PCI trend. Additionally, to evaluate the long-term outcomes of the NSTEMI and STEMI groups, we excluded patients who died in-hospital (Figure 1).
Patients with NSTEMI tend to be older and have a lower rate of acute revascularization than those with STEMI [34]. In our study, in both the DM and non-DM groups, the patients in the NSTEMI group had a higher mean age than those in the STEMI group (65.5 ± 11.3 years vs. 63.0 ± 11.9, p <0.001 and 63.5 ± 12.6 years vs. 61.4 ± 13.0, p <0.001, respectively) and the total number of patients who underwent PCI within 24 h lower in the NSTEMI group than that in the STEMI group (85.0% vs. 96.5%, p <0.001 and 87.5% vs. 96.7%, p <0.001, respectively, Table 1). In the Okura et al. report, non-CD (15.1% vs. 8.4%, p <0.001, respectively) rate was significantly higher in the NSTEMI group than that in the STEMI group during a median 4.3-year follow-up period [34]. Recently, in a study of a CKD population [35], the non-CD (aHR, 1.960; p = 0.004) rate was significantly higher in the NSTEMI group than in the STEMI group. In our study, after both multivariable-adjusted (aHR, 2.223; p = 0.006) and PS-adjusted (aHR, 2.484; p = 0.004) analyses, the non-CD rate was significantly higher in the DM group than that in the STEMI group (Table 2). Furthermore, after multivariable-adjusted analysis, the non-CD rate in the NSTEMI group was significantly higher in the DM group than that in the non-DM group (aHR, 2.810; p <0.001, Table 3). The main causes of non-CD in the DM group were multiple organ failure (0.7% vs. 0.2%, p = 0.008) and CVA (0.9% vs. 0.4%, p = 0.034) (Supplementary material 4). In a previous report [36], CVA was independently predicted by DM (HR, 1.43; 95% CI, 1.04-1.97; p = 0.03) and associated with a significantly increased 3-year mortality rate (aHR, 2.39; p = 0.004). Hence, DM accelerates atherosclerosis in multiple vascular beds and causes severe coronary atherosclerosis. Coronary and cerebrovascular diseases frequently coexist because of their similar pathogeneses [37]. In our study, in the non-DM group, the mortality rate (all-cause and CD) was significantly higher in the NSTEMI group than in the STEMI group. However, in the DM group, the mortality rate was not significantly different between the two MI groups (Table 2). This result may be related to the relatively higher CD rate in the DM and STEMI groups than that in the non-DM and STEMI groups (aHR, 2.534; p <0.001, Table 3). Additionally, this result may be related to the insignificantly different non-CD rates in the DM and STEMI groups compared to the non-DM and STEMI groups (aHR, 1.309; p = 0.376, Table 3).
The incidence of recurrent ischemic events after AMI is higher in the first year; in subsequent years, it is based on a number of cardiovascular risk factors [38]. Recently, Kim et al. [39] reported that, in patients with AMI, the cumulative incidence of re-MI was significantly higher in the DM group than in the normoglycemia group (aHR, 1.752; 95% CI, 1.087–2.823; p = 0.021). In our study, in all patients, the re-MI rate was significantly higher in the DM group than in the non-DM group (aHR, 1.479; 95% CI, 1.131–1.934; p = 0.004, Table 3). As shown in Figure 2, the re-MI rate increased continuously in the DM group, regardless of AMI type. In a Danish study [40], the risk of ST (definite, probable, or possible) did not differ significantly between the DM and non-DM groups in patients with STEMI (aHR, 1.50; 95% CI, 0.92–2.45). Recently, a subgroup analysis of the ultrathin strut biodegradable polymer sirolimus-eluting stent versus durable polymer everolimus-eluting stent for percutaneous coronary revascularization (BIOSCIENCE) trial [41] showed that the 5-year ST (definite or probable) was significantly higher in the DM group than in the non-DM group (rate ratio [RR], 2.05; 95% CI, 1.45–2.90; p <0.001) in all patients after new-generation DES implantation. Furthermore, in their study, the target lesion failure (RR, 1.87; p <0.001) and target vessel failure (RR, 1.76; p <0.001) rates were significantly higher in the DM group than in the non-DM group [41]. In our study, although patients in the DM with STEMI group showed comparable ST (definite or probable) rates to those with patients in the non-DM STEMI group (aHR, 1.394; 95% CI, 0.803–2.419; p = 0.237; Table 3), the overall ST rate was significantly higher in the DM group than in the non-DM group (aHR, 1.641; 95% CI, 1.050-2.564; p = 0.030, Table 3). Additionally, in overall patients, any repeat revascularization rate was significantly higher in the DM group than in the non-DM group (aHR, 1.310; 95% CI, 1.069–1.605; p = 0.009). In our study, reduced LVEF, cardiogenic shock, CPR on admission, peak troponin-I, NT-ProBNP, and lipid-lowering agents were common independent predictors of MACE in both the DM and non-DM groups (Supplementary material 5). These independent predictors are well-known predictors of MACE in patients with AMI [33,42,43].
Many previous studies have demonstrated that a higher long-term mortality was observed in the NSTEMI group than that in the STEMI group regardless of the type of death [7,9,10,31,44]. Consistent with the results of previous studies [7,9,10,31,44], in all patients in our study, the rates of MACE, all-cause death, CD, and non-CD were significantly higher in the NSTEMI group than in the STEMI group. However, the higher non-CD rate in the NSTEMI group was more evident in the DM group, and the higher CD rate in the NSTEMI group was more evident in the non-DM group. Hence, we believe that strategies for reducing the non-CD rate in patients with DM and those for reducing the CD rate in patients without DM could be beneficial for the NSTEMI group after successful PCI with new-generation DES. Although the study population was insufficient to show meaningful results, more than 50 high-volume universities or community hospitals with facilities for primary PCI and on-site cardiac surgery participated in this study. Regarding the relatively higher incidence of multiple organ failure and CVA in the DM and NSTEMI groups than in the non-DM and STEMI groups (Supplementary material 4), more well-established and regular follow-ups [9] and more focused and diverse secondary prevention therapies [31], including those involving lipid-lowering agents (Supplementary materials 5 and 6), are required to reduce the occurrence of non-CD in patients with NSTEMI. Therefore, we believe that the results of this comparative study could provide interventional cardiologists with meaningful information regarding the treatment strategies for patients with two different types of AMI according to the presence or absence of DM.
This study has other limitations. First, because we used registry data, there may have been some under-reported and/or missing data. Second, this study was based on discharge medications since we could not precisely determine the participants’ adherence or non-adherence to their antidiabetic drugs during the 2-year follow-up period. Third, although the interval from symptom onset to PCI is an important determinant of major clinical outcomes, this variable included many missing values in the registry data. Therefore, we could not include this variable in this study. Fourth, although we performed multivariable-adjusted and PS-adjusted analyses to strengthen our results, variables not included in the KAMIR may have affected the study outcomes. Fifth, it is not certain that our population had 100% T2DM, just based on the age of discovery of their diabetes. Occasionally, T1DM can occur in individuals aged over 30 years [45]. Moreover, there were some missing values concerning patient-reported history, such as the presence or absence of a history of ketoacidosis and other medical records indicating T2DM in the KAMIR data due to the registry nature of this study. These factors may be important limitations of this study. Sixth, the progression of atherosclerosis and the development of cardiovascular events can be altered by various recently developed antidiabetic agents (e.g., sodium-glucose cotransporter 2 inhibitor, glucagon-like Peptide-1 [GLP-1] agonist) [46]. Unfortunately, we could not obtain information about the various recently developed antidiabetic agents that had been prescribed from the KAMIR registry. This is another important limitation of this study. Seventh, this retrospective study was a long-term (November 2005 to June 2015) study of patients with AMI, which could have affected the clinical outcomes. Finally, the 2-year follow-up period of this study was relatively short and may have been inadequate for determining long-term major clinical outcomes.
After,
- Discussion
The main findings of this study were as follows: (1) in the DM group, although MACE, all-cause death, CD, recurrent MI, any repeat revascularization, and ST rates were not significantly different between the NSTEMI and STEMI groups, the non-CD rate was significantly higher in the NSTEMI group than in the STEMI group; (2) in the non-DM group, the MACE, all-cause death, and CD rates were significantly higher in the NSTEMI group than in the STEMI group; (3) in the NSTEMI group, the MACE, all-cause death, non-CD, and ST rates were significantly higher in the DM group than in the non-DM group; (4) in the STEMI group, the MACE, all-cause death, CD, recurrent MI, and any repeat revascularization rates were significantly higher in the DM group than in the non-DM group; and (5) reduced LVEF (<40%), cardiogenic shock, CPR on admission, peak troponin-I level, NT-ProBNP level, and lipid-lowering agents were common independent predictors for MACE in both the DM and non-DM groups.
STEMI is the result of acute occlusion of the IRA and is associated with transmural ischemia, whereas NSTEMI is caused by transient or incomplete coronary artery occlusion, resulting in non-transmural subendocardial ischemia [4]. In previous studies [9,31], the 6-month post-discharge mortality (6.2% vs. 4.8%, respectively) and 1-year mortality (11.6% vs. 9.0%, respectively) rates were higher in the NSTEMI group than in the STEMI group. Although these two randomized studies [9,31] are valuable for estimating comparative clinical outcomes between NSTEMI and STEMI groups, they [9,31] were conducted before the new-generation DES era and were not limited to patients with DM. Hyperglycemia contributes to increased mortality and morbidity rates through an oxidative-linked mechanism [32] and may exert significant hemodynamic effects even in normal study participants [12]. Furthermore, hyperglycemia caused by oxidative stress, inflammation, apoptosis, endothelial dysfunction, hypercoagulation, and platelet aggregation [33] could damage the ischemic myocardium in patients with NSTEMI [12]. In a study by Hao et al. [12], among 890 patients with NSTEMI who underwent PCI, admission hyperglycemia was an independent predictor of a 30-day (aHR, 1.014; p < 0.001 and aHR, 1.018, p < 0.001, respectively) and 3-year MACE (aHR, 1.009; p < 0.001 and aHR, 1.017, p < 0.001, respectively) in patients with and without DM. Recently, Li et al. [13] demonstrated that the incidences of all-cause death (1.1%) and MACE (3.4%) were significantly lower in patients without a history of DM and an HbA1c level of <6.5% at admission. However, DM patients with poor glycemic control at admission experienced high rates of all-cause death (18.8%) and MACE (25%) in 350 consecutive patients with STEMI during a 2-year follow-up period. Similarly, in our study, both in the NSTEMI and STEMI groups, the rates of all-cause death (aHR, 1.755; p = 0.001 and aHR; 2.040, p < 0.001, respectively) and MACE (aHR, 1.328; p = 0.006 and aHR, 1.527; p < 0.001, respectively) were significantly higher in the DM group than in the non-DM group (Table 3). However, the study population in these studies [12,13] was limited to patients with NSTEMI or STEMI. DESs have been developed to overcome the limitations of BMS deployment, such as neointimal hyperplasia and repeat revascularization [22,24]. In the era of DES, second-generation DES (2G-DES) is the most commonly used DES because it can solve the problems of 1G-DES, such as inflammation and restenosis, and decrease the mortality rate (aHR, 1.534; p = 0.009) [14]. Hence, to provide more meaningful results and compensate for the shortcomings of the previous studies [9,12,13,31], we compared the 2-year clinical outcomes between the NSTEMI and STEMI groups according to the presence or absence of DM. The study population was strictly confined to patients with AMI who underwent successful implantation of a new-generation DES to reflect the current PCI trend. Additionally, to evaluate the long-term outcomes of the NSTEMI and STEMI groups, we excluded patients who died in the hospital (Figure 1).
Patients with NSTEMI tend to be older and have a lower rate of acute revascularization than those with STEMI [34]. In our study, in both the DM and non-DM groups, the patients in the NSTEMI group had a higher mean age than those in the STEMI group (65.5 ± 11.3 years vs. 63.0 ± 11.9, p < 0.001 and 63.5 ± 12.6 years vs. 61.4 ± 13.0, p < 0.001, respectively) and the total number of patients who underwent PCI within 24 h was lower in the NSTEMI group than that in the STEMI group (85.0% vs. 96.5%, p < 0.001 and 87.5% vs. 96.7%, p < 0.001, respectively, Table 1). Although this study did not include in-hospital outcomes, it could be stated that STEMI is a higher-risk disease in the acute phase [6,7], for which we have gathered treatment knowledge, while NSTEMI is a higher-risk disease in the chronic phase owing to the higher complexity of these patients [9,10]. Okura et al. reported that the non-CD rate (15.1% vs. 8.4 %, p < 0.001) was significantly higher in the NSTEMI group than in the STEMI group during a median 4.3-year follow-up period [34]. In a recent study involving patients with chronic kidney disease [36], the non-CD rate (aHR, 1.960; p = 0.004) was significantly higher in the NSTEMI group than in the STEMI group. In our study, after both multivariable- (aHR, 2.223; p = 0.006) and PS-adjusted (aHR, 2.484; p = 0.004) analyses, the non-CD rate was significantly higher in the DM group than that in the STEMI group (Table 2). Furthermore, after multivariable-adjusted analysis, the non-CD rate in the NSTEMI group was significantly higher in the DM group than that in the non-DM group (aHR, 2.810; p < 0.001, Table 3). The main causes of non-CD in the DM group were multiple organ failure (0.7% vs. 0.2%, p = 0.008) and CVA (0.9% vs. 0.4%, p = 0.034) (Supplementary Material 4). In a previous report [37], CVA was independently predicted by DM (HR, 1.43; 95% CI, 1.04-1.97; p = 0.03) and was associated with a significantly increased 3-year mortality rate (aHR, 2.39; p = 0.004). DM accelerates atherosclerosis in multiple vascular beds and causes severe coronary atherosclerosis. Coronary and cerebrovascular diseases frequently coexist because of their similar pathogeneses [38]. In our study, in the non-DM group, the mortality rate (all-cause and CD) was significantly higher in the NSTEMI group than in the STEMI group; however, in the DM group, the mortality rate did not significantly differ between the two MI groups (Table 2). This result may be related to the relatively higher CD rate in the DM and STEMI groups than that in the non-DM and STEMI groups (aHR, 2.534; p < 0.001, Table 3). Additionally, this result may be related to the insignificantly different non-CD rates in the DM and STEMI groups compared to those in the non-DM and STEMI groups (aHR, 1.309; p = 0.376, Table 3).
The incidence of recurrent ischemic events after AMI is higher in the first year, and, in subsequent years, it is based on several cardiovascular risk factors [39]. Recently, Kim et al. [40] reported that in patients with AMI, the cumulative incidence of re-MI was significantly higher in the DM group than in the normoglycemia group (aHR, 1.752; 95% CI, 1.087–2.823; p = 0.021). Among all the patients in our study, the re-MI rate was significantly higher in the DM group than in the non-DM group (aHR, 1.479; 95% CI, 1.131–1.934; p = 0.004, Table 3). The re-MI rate increased continuously in the DM group, regardless of AMI type (Figure 2). In a Danish study [41], the risk of ST (definite, probable, or possible) did not differ significantly between DM and non-DM groups in patients with STEMI (aHR, 1.50; 95% CI, 0.92–2.45). Recently, a subgroup analysis of the ultrathin strut biodegradable polymer sirolimus-eluting stent versus the durable polymer everolimus-eluting stent for percutaneous coronary revascularization trial [42] showed that the 5-year ST (definite or probable) was significantly higher in the DM group than in the non-DM group (rate ratio [RR], 2.05; 95% CI, 1.45–2.90; p < 0.001) in all patients after new-generation DES implantation. Furthermore, in their study, the target lesion failure (rate ratio, 1.87; p < 0.001) and target vessel failure (rate ratio, 1.76; p < 0.001) rates were significantly higher in the DM group than in the non-DM group [42]. In our study, although patients in the DM with STEMI group showed comparable ST (definite or probable) rates to those in the non-DM STEMI group (aHR, 1.394; 95% CI, 0.803–2.419; p = 0.237; Table 3), the overall ST rate was significantly higher in the DM group than in the non-DM group (aHR, 1.641; 95% CI, 1.050-2.564; p = 0.030, Table 3). Additionally, the repeat revascularization rate was significantly higher in the DM group than in the non-DM group (aHR, 1.310; 95% CI, 1.069–1.605; p = 0.009). In our study, reduced LVEF, cardiogenic shock, CPR on admission, peak troponin-I, NT-ProBNP, and lipid-lowering agents were common independent predictors of MACE in both the DM and non-DM groups (Supplementary Material 5). These are well-known independent predictors of MACE in patients with AMI [33,43,44].
Several previous studies have demonstrated that higher long-term mortality was observed in the NSTEMI group than in the STEMI group, regardless of the type of death [7,9,10,31,45]. Consistent with the results of previous studies [7,9,10,31,45], in all the patients in our study, the rates of MACE, all-cause death, CD, and non-CD were significantly higher in the NSTEMI group than in the STEMI group. However, the higher non-CD rate in the NSTEMI group was more evident in the DM group, while the higher CD rate in the NSTEMI group was more evident in the non-DM group. Hence, we believe that strategies for reducing the non-CD rate in patients with DM and those for reducing the CD rate in patients without DM could be beneficial for the NSTEMI group after successful PCI with new-generation DES. Although the sample size of the study population was too small to demonstrate meaningful results, more than 50 high-volume universities and community hospitals with facilities for primary PCI and on-site cardiac surgery participated in this study. Regarding the relatively higher incidence of multiple organ failure and CVA in the DM and NSTEMI groups than in the non-DM and STEMI groups (Supplementary Material 4), more well-established and regular follow-ups [9] as well as more focused and diverse secondary prevention therapies [31], including those involving lipid-lowering agents (Supplementary Materials 5 and 6), are required to reduce the occurrence of non-CD in patients with NSTEMI. Therefore, we believe that the results of this comparative study could provide interventional cardiologists with meaningful information regarding treatment strategies for patients with two different types of AMI according to the presence or absence of DM.
This study has several limitations. First, because we used registry data, there may have been some under-reported or missing data. Second, this study was based on discharge medications because we could not precisely determine the participants’ adherence or non-adherence to their antidiabetic drugs during the 2-year follow-up period. Third, although the interval from symptom onset to PCI was an important determinant of major clinical outcomes, this variable included many missing values in the registry data; therefore, we could not include this variable in our study. Fourth, although we performed multivariable- and PS-adjusted analyses to strengthen our results, variables not included in the KAMIR may have affected the study outcomes. Fifth, it is not certain that our population had 100% T2DM based only on the age at which diabetes was discovered. Occasionally, type 1 DM occurs in individuals aged over 30 years [48]. Moreover, there were some missing values concerning patient-reported history, such as the presence or absence of a history of ketoacidosis and other medical records indicating T2DM in the KAMIR data, owing to the registry-based nature of this study. These factors may be considered as critical limitations of this study. Sixth, this retrospective study was a long-term (November 2005 to June 2015) study of patients with AMI, which could have affected clinical outcomes. Finally, the 2-year follow-up period in this study was relatively short and may have been inadequate for determining long-term major clinical outcomes.
- Authors provided values of glycosilated hemoglobin. HA1c (not at a target level in both groups) was higher in STEMI patients, while insulin treatment was higher in NSTEMI group. Is it possible that NSTEMI patients had, at the time of the MI, a more decompensated diabetes? If yes, could have influenced the non CV death among the 2 groups?
Answer: We sincerely thank you for reviewer’s valuable comments and question. Yes, regarding above baseline characteristics of the study population, we also think that the number of patients with decompensated diabetes could be higher in the NSTEMI than in the STEMI. Therefore, we added this content in the “Discussion” section as follows,
Before,
- Discussion
The main findings of this study were as follows: (1) in the DM group, although MACE, all-cause death, CD, recurrent MI, any repeat revascularization, and ST rates were not significantly different between the NSTEMI and STEMI groups, the non-CD rate was significantly higher in the NSTEMI group than in the STEMI group; (2) in the non-DM group, the MACE, all-cause death, and CD rates were significantly higher in the NSTEMI group than in the STEMI group; (3) in the NSTEMI group, the MACE, all-cause death, non-CD, and ST rates were significantly higher in the DM group than in the non-DM group; (4) in the STEMI group, the MACE, all-cause death, CD, recurrent MI, and any repeat revascularization rates were significantly higher in the DM group than in the non-DM group; and (5) reduced LVEF (<40%), cardiogenic shock, CPR on admission, peak troponin-I level, NT-ProBNP level, and lipid-lowering agents were common independent predictors for MACE in both the DM and non-DM groups.
STEMI is the result of acute occlusion of the IRA and is associated with transmural ischemia, whereas NSTEMI is caused by transient or incomplete coronary artery occlusion, resulting in non-transmural subendocardial ischemia [4]. In the previous studies [9,31], the 6-month after post-discharge mortality (6.2% vs. 4.8%, respectively) and 1-year mortality (11.6% vs. 9.0%, respectively) were higher in the NSTEMI group than that in the STEMI group. Although these two randomized studies [9,31] are valuable for estimating comparative clinical outcomes between the NSTEMI and STEMI groups, they [9,31] were conducted before the new-generation DES era and were not limited to patients with DM. Hyperglycemia contributes to increased mortality and morbidity rates through an oxidative-linked mechanism [32] and may exert significant hemodynamic effects even in normal subjects [12]. Furthermore, hyperglycemia caused by oxidative stress, inflammation, apoptosis, endothelial dysfunction, hypercoagulation, and platelet aggregation [33] could damage the ischemic myocardium in patients with NSTEMI [12]. In a study by Hao et al. [12], among 890 NSTEMI patients who underwent PCI, admission hyperglycemia was an independent predictor of 30-day (aHR, 1.014; p <0.001 and aHR, 1.018, p <0.001, respectively) and 3-year MACE (aHR, 1.009; p <0.001 and aHR, 1.017, p <0.001, respectively) in patients with and without DM. Recently, Li et al. [13] demonstrated that the incidence of all-cause death (1.1%) and MACE (3.4%) was significantly the low in patients without a DM history and admission HbA1c level < 6.5%. However, DM patients with poor glycemic control on admission experienced the high rates of all-cause death (18.8%) and MACE (25%) in their 350 consecutive patients with STEMI during a 2-year follow-up period. Similarly, in our study, both in the NSTEMI group and in the STEMI group, the rates of all-cause death (aHR, 1.755; p = 0.001 and aHR; 2.040, p <0.001, respectively) and MACE (aHR, 1.328; p = 0.006 and aHR, 1.527; p <0.001, respectively) were significantly higher in the DM group than in the non-DM group (Table 3). However, the study population in these studies [12,13] was limited to patients with NSTEMI or STEMI, separately. DESs were developed to overcome the limitations of BMS deployment, such as neointimal hyperplasia and repeat revascularization [22,24]. In the era of DES, second-generation DES (2G-DES) is the most commonly used DES because it can solve the problems of 1G-DES, such as inflammation and restenosis, and can decrease the mortality rate (aHR, 1.534; p = 0.009) [14]. Hence, to provide more meaningful results and compensate for the shortcomings of the previous studies [9,12,13,31] from the current study, we compared the 2-year clinical outcomes between the NSTEMI and STEMI groups according to the presence or absence of DM. The study population was strictly confined to patients with AMI who underwent successful implantation of new-generation DES to reflect the current PCI trend. Additionally, to evaluate the long-term outcomes of the NSTEMI and STEMI groups, we excluded patients who died in-hospital (Figure 1).
Patients with NSTEMI tend to be older and have a lower rate of acute revascularization than those with STEMI [34]. In our study, in both the DM and non-DM groups, the patients in the NSTEMI group had a higher mean age than those in the STEMI group (65.5 ± 11.3 years vs. 63.0 ± 11.9, p <0.001 and 63.5 ± 12.6 years vs. 61.4 ± 13.0, p <0.001, respectively) and the total number of patients who underwent PCI within 24 h lower in the NSTEMI group than that in the STEMI group (85.0% vs. 96.5%, p <0.001 and 87.5% vs. 96.7%, p <0.001, respectively, Table 1). Although this study didn’t include in-hospital outcomes, STEMI is a more dangerous disease in the acute phase [6,7], that we have learnt how to treat, while NSTEMI is a more dangerous disease in the chronic phase, due to a higher complexity of these patients [9,10]. In the Okura et al. report, non-CD (15.1% vs. 8.4%, p <0.001, respectively) rate was significantly higher in the NSTEMI group than that in the STEMI group during a median 4.3-year follow-up period [34]. Recently, in a study of a CKD population [35], the non-CD (aHR, 1.960; p = 0.004) rate was significantly higher in the NSTEMI group than in the STEMI group. In our study, after both multivariable-adjusted (aHR, 2.223; p = 0.006) and PS-adjusted (aHR, 2.484; p = 0.004) analyses, the non-CD rate was significantly higher in the DM group than that in the STEMI group (Table 2). Furthermore, after multivariable-adjusted analysis, the non-CD rate in the NSTEMI group was significantly higher in the DM group than that in the non-DM group (aHR, 2.810; p <0.001, Table 3). The main causes of non-CD in the DM group were multiple organ failure (0.7% vs. 0.2%, p = 0.008) and CVA (0.9% vs. 0.4%, p = 0.034) (Supplementary material 4). In a previous report [36], CVA was independently predicted by DM (HR, 1.43; 95% CI, 1.04-1.97; p = 0.03) and associated with a significantly increased 3-year mortality rate (aHR, 2.39; p = 0.004). Hence, DM accelerates atherosclerosis in multiple vascular beds and causes severe coronary atherosclerosis. Coronary and cerebrovascular diseases frequently coexist because of their similar pathogeneses [37]. In our study, in the non-DM group, the mortality rate (all-cause and CD) was significantly higher in the NSTEMI group than in the STEMI group. However, in the DM group, the mortality rate was not significantly different between the two MI groups (Table 2). This result may be related to the relatively higher CD rate in the DM and STEMI groups than that in the non-DM and STEMI groups (aHR, 2.534; p <0.001, Table 3). Additionally, this result may be related to the insignificantly different non-CD rates in the DM and STEMI groups compared to the non-DM and STEMI groups (aHR, 1.309; p = 0.376, Table 3).
The incidence of recurrent ischemic events after AMI is higher in the first year; in subsequent years, it is based on a number of cardiovascular risk factors [38]. Recently, Kim et al. [39] reported that, in patients with AMI, the cumulative incidence of re-MI was significantly higher in the DM group than in the normoglycemia group (aHR, 1.752; 95% CI, 1.087–2.823; p = 0.021). In our study, in all patients, the re-MI rate was significantly higher in the DM group than in the non-DM group (aHR, 1.479; 95% CI, 1.131–1.934; p = 0.004, Table 3). As shown in Figure 2, the re-MI rate increased continuously in the DM group, regardless of AMI type. In a Danish study [40], the risk of ST (definite, probable, or possible) did not differ significantly between the DM and non-DM groups in patients with STEMI (aHR, 1.50; 95% CI, 0.92–2.45). Recently, a subgroup analysis of the ultrathin strut biodegradable polymer sirolimus-eluting stent versus durable polymer everolimus-eluting stent for percutaneous coronary revascularization (BIOSCIENCE) trial [41] showed that the 5-year ST (definite or probable) was significantly higher in the DM group than in the non-DM group (rate ratio [RR], 2.05; 95% CI, 1.45–2.90; p <0.001) in all patients after new-generation DES implantation. Furthermore, in their study, the target lesion failure (rate ratio, 1.87; p <0.001) and target vessel failure (rate ratio, 1.76; p <0.001) rates were significantly higher in the DM group than in the non-DM group [41]. In our study, although patients in the DM with STEMI group showed comparable ST (definite or probable) rates to those with patients in the non-DM STEMI group (aHR, 1.394; 95% CI, 0.803–2.419; p = 0.237; Table 3), the overall ST rate was significantly higher in the DM group than in the non-DM group (aHR, 1.641; 95% CI, 1.050-2.564; p = 0.030, Table 3). Additionally, in overall patients, any repeat revascularization rate was significantly higher in the DM group than in the non-DM group (aHR, 1.310; 95% CI, 1.069–1.605; p = 0.009). In our study, reduced LVEF, cardiogenic shock, CPR on admission, peak troponin-I, NT-ProBNP, and lipid-lowering agents were common independent predictors of MACE in both the DM and non-DM groups (Supplementary material 5). These independent predictors are well-known predictors of MACE in patients with AMI [33,42,43].
Many previous studies have demonstrated that a higher long-term mortality was observed in the NSTEMI group than that in the STEMI group regardless of the type of death [7,9,10,31,44]. Consistent with the results of previous studies [7,9,10,31,44], in all patients in our study, the rates of MACE, all-cause death, CD, and non-CD were significantly higher in the NSTEMI group than in the STEMI group. However, the higher non-CD rate in the NSTEMI group was more evident in the DM group, and the higher CD rate in the NSTEMI group was more evident in the non-DM group. Hence, we believe that strategies for reducing the non-CD rate in patients with DM and those for reducing the CD rate in patients without DM could be beneficial for the NSTEMI group after successful PCI with new-generation DES. Although the study population was insufficient to show meaningful results, more than 50 high-volume universities or community hospitals with facilities for primary PCI and on-site cardiac surgery participated in this study. Regarding the relatively higher incidence of multiple organ failure and CVA in the DM and NSTEMI groups than in the non-DM and STEMI groups (Supplementary material 4), more well-established and regular follow-ups [9] and more focused and diverse secondary prevention therapies [31], including those involving lipid-lowering agents (Supplementary materials 5 and 6), are required to reduce the occurrence of non-CD in patients with NSTEMI. Therefore, we believe that the results of this comparative study could provide interventional cardiologists with meaningful information regarding the treatment strategies for patients with two different types of AMI according to the presence or absence of DM.
This study has other limitations. First, because we used registry data, there may have been some under-reported and/or missing data. Second, this study was based on discharge medications since we could not precisely determine the participants’ adherence or non-adherence to their antidiabetic drugs during the 2-year follow-up period. Third, although the interval from symptom onset to PCI is an important determinant of major clinical outcomes, this variable included many missing values in the registry data. Therefore, we could not include this variable in this study. Fourth, although we performed multivariable-adjusted and PS-adjusted analyses to strengthen our results, variables not included in the KAMIR may have affected the study outcomes. Fifth, it is not certain that our population had 100% T2DM, just based on the age of discovery of their diabetes. Occasionally, type 1 DM can occur in individuals aged over 30 years [45]. Moreover, there were some missing values concerning patient-reported history, such as the presence or absence of a history of ketoacidosis and other medical records indicating T2DM in the KAMIR data due to the registry nature of this study. These factors may be important limitations of this study. Sixth, the progression of atherosclerosis and the development of cardiovascular events can be altered by various recently developed antidiabetic agents (e.g., sodium-glucose cotransporter 2 inhibitor, glucagon-like Peptide-1 [GLP-1] agonist) [46]. Unfortunately, we could not obtain information about the various recently developed antidiabetic agents that had been prescribed from the KAMIR registry. This is another important limitation of this study. Seventh, this retrospective study was a long-term (November 2005 to June 2015) study of patients with AMI, which could have affected the clinical outcomes. Finally, the 2-year follow-up period of this study was relatively short and may have been inadequate for determining long-term major clinical outcomes.
After,
- Discussion
The main findings of this study were as follows: (1) in the DM group, although MACE, all-cause death, CD, recurrent MI, any repeat revascularization, and ST rates were not significantly different between the NSTEMI and STEMI groups, the non-CD rate was significantly higher in the NSTEMI group than in the STEMI group; (2) in the non-DM group, the MACE, all-cause death, and CD rates were significantly higher in the NSTEMI group than in the STEMI group; (3) in the NSTEMI group, the MACE, all-cause death, non-CD, and ST rates were significantly higher in the DM group than in the non-DM group; (4) in the STEMI group, the MACE, all-cause death, CD, recurrent MI, and any repeat revascularization rates were significantly higher in the DM group than in the non-DM group; and (5) reduced LVEF (<40%), cardiogenic shock, CPR on admission, peak troponin-I level, NT-ProBNP level, and lipid-lowering agents were common independent predictors for MACE in both the DM and non-DM groups.
STEMI is the result of acute occlusion of the IRA and is associated with transmural ischemia, whereas NSTEMI is caused by transient or incomplete coronary artery occlusion, resulting in non-transmural subendocardial ischemia [4]. In previous studies [9,31], the 6-month post-discharge mortality (6.2% vs. 4.8%, respectively) and 1-year mortality (11.6% vs. 9.0%, respectively) rates were higher in the NSTEMI group than in the STEMI group. Although these two randomized studies [9,31] are valuable for estimating comparative clinical outcomes between NSTEMI and STEMI groups, they [9,31] were conducted before the new-generation DES era and were not limited to patients with DM. Hyperglycemia contributes to increased mortality and morbidity rates through an oxidative-linked mechanism [32] and may exert significant hemodynamic effects even in normal study participants [12]. Furthermore, hyperglycemia caused by oxidative stress, inflammation, apoptosis, endothelial dysfunction, hypercoagulation, and platelet aggregation [33] could damage the ischemic myocardium in patients with NSTEMI [12]. In a study by Hao et al. [12], among 890 patients with NSTEMI who underwent PCI, admission hyperglycemia was an independent predictor of a 30-day (aHR, 1.014; p < 0.001 and aHR, 1.018, p < 0.001, respectively) and 3-year MACE (aHR, 1.009; p < 0.001 and aHR, 1.017, p < 0.001, respectively) in patients with and without DM. Recently, Li et al. [13] demonstrated that the incidences of all-cause death (1.1%) and MACE (3.4%) were significantly lower in patients without a history of DM and an HbA1c level of <6.5% at admission. However, DM patients with poor glycemic control at admission experienced high rates of all-cause death (18.8%) and MACE (25%) in 350 consecutive patients with STEMI during a 2-year follow-up period. Similarly, in our study, both in the NSTEMI and STEMI groups, the rates of all-cause death (aHR, 1.755; p = 0.001 and aHR; 2.040, p < 0.001, respectively) and MACE (aHR, 1.328; p = 0.006 and aHR, 1.527; p < 0.001, respectively) were significantly higher in the DM group than in the non-DM group (Table 3). However, the study population in these studies [12,13] was limited to patients with NSTEMI or STEMI. DESs have been developed to overcome the limitations of BMS deployment, such as neointimal hyperplasia and repeat revascularization [22,24]. In the era of DES, second-generation DES (2G-DES) is the most commonly used DES because it can solve the problems of 1G-DES, such as inflammation and restenosis, and decrease the mortality rate (aHR, 1.534; p = 0.009) [14]. Hence, to provide more meaningful results and compensate for the shortcomings of the previous studies [9,12,13,31], we compared the 2-year clinical outcomes between the NSTEMI and STEMI groups according to the presence or absence of DM. The study population was strictly confined to patients with AMI who underwent successful implantation of a new-generation DES to reflect the current PCI trend. Additionally, to evaluate the long-term outcomes of the NSTEMI and STEMI groups, we excluded patients who died in the hospital (Figure 1).
Patients with NSTEMI tend to be older and have a lower rate of acute revascularization than those with STEMI [34]. In our study, in both the DM and non-DM groups, the patients in the NSTEMI group had a higher mean age than those in the STEMI group (65.5 ± 11.3 years vs. 63.0 ± 11.9, p < 0.001 and 63.5 ± 12.6 years vs. 61.4 ± 13.0, p < 0.001, respectively) and the total number of patients who underwent PCI within 24 h was lower in the NSTEMI group than that in the STEMI group (85.0% vs. 96.5%, p < 0.001 and 87.5% vs. 96.7%, p < 0.001, respectively, Table 1). Although this study did not include in-hospital outcomes, it could be stated that STEMI is a higher-risk disease in the acute phase [6,7], for which we have gathered treatment knowledge, while NSTEMI is a higher-risk disease in the chronic phase owing to the higher complexity of these patients [9,10]. Moreover, in our study, although HbA1c levels were higher in patients with STEMI, the number of patients undergoing insulin treatment was higher in the NSTEMI group. Patients with NSTEMI may have had greater decompensated diabetes at the time of MI. Insulin-treated patients with DM were associated with significantly higher short- and long-term adverse cardiovascular outcomes after PCI than those not treated with insulin therapy [35]. However, long-term comparative results between patients with insulin-treated DM and non-insulin-treated DM after NSTEMI and STEMI are very limited. Further studies are required to evaluate long-term clinical outcomes in the NSTEMI and STEMI groups. Okura et al. reported that the non-CD rate (15.1% vs. 8.4 %, p < 0.001) was significantly higher in the NSTEMI group than in the STEMI group during a median 4.3-year follow-up period [34]. In a recent study involving patients with chronic kidney disease [36], the non-CD rate (aHR, 1.960; p = 0.004) was significantly higher in the NSTEMI group than in the STEMI group. In our study, after both multivariable- (aHR, 2.223; p = 0.006) and PS-adjusted (aHR, 2.484; p = 0.004) analyses, the non-CD rate was significantly higher in the DM group than that in the STEMI group (Table 2). Furthermore, after multivariable-adjusted analysis, the non-CD rate in the NSTEMI group was significantly higher in the DM group than that in the non-DM group (aHR, 2.810; p < 0.001, Table 3). The main causes of non-CD in the DM group were multiple organ failure (0.7% vs. 0.2%, p = 0.008) and CVA (0.9% vs. 0.4%, p = 0.034) (Supplementary Material 4). In a previous report [37], CVA was independently predicted by DM (HR, 1.43; 95% CI, 1.04-1.97; p = 0.03) and was associated with a significantly increased 3-year mortality rate (aHR, 2.39; p = 0.004). DM accelerates atherosclerosis in multiple vascular beds and causes severe coronary atherosclerosis. Coronary and cerebrovascular diseases frequently coexist because of their similar pathogeneses [38]. In our study, in the non-DM group, the mortality rate (all-cause and CD) was significantly higher in the NSTEMI group than in the STEMI group; however, in the DM group, the mortality rate did not significantly differ between the two MI groups (Table 2). This result may be related to the relatively higher CD rate in the DM and STEMI groups than that in the non-DM and STEMI groups (aHR, 2.534; p < 0.001, Table 3). Additionally, this result may be related to the insignificantly different non-CD rates in the DM and STEMI groups compared to those in the non-DM and STEMI groups (aHR, 1.309; p = 0.376, Table 3).
The incidence of recurrent ischemic events after AMI is higher in the first year, and, in subsequent years, it is based on several cardiovascular risk factors [39]. Recently, Kim et al. [40] reported that in patients with AMI, the cumulative incidence of re-MI was significantly higher in the DM group than in the normoglycemia group (aHR, 1.752; 95% CI, 1.087–2.823; p = 0.021). Among all the patients in our study, the re-MI rate was significantly higher in the DM group than in the non-DM group (aHR, 1.479; 95% CI, 1.131–1.934; p = 0.004, Table 3). The re-MI rate increased continuously in the DM group, regardless of AMI type (Figure 2). In a Danish study [41], the risk of ST (definite, probable, or possible) did not differ significantly between DM and non-DM groups in patients with STEMI (aHR, 1.50; 95% CI, 0.92–2.45). Recently, a subgroup analysis of the ultrathin strut biodegradable polymer sirolimus-eluting stent versus the durable polymer everolimus-eluting stent for percutaneous coronary revascularization trial [42] showed that the 5-year ST (definite or probable) was significantly higher in the DM group than in the non-DM group (rate ratio [RR], 2.05; 95% CI, 1.45–2.90; p < 0.001) in all patients after new-generation DES implantation. Furthermore, in their study, the target lesion failure (rate ratio, 1.87; p < 0.001) and target vessel failure (rate ratio, 1.76; p < 0.001) rates were significantly higher in the DM group than in the non-DM group [42]. In our study, although patients in the DM with STEMI group showed comparable ST (definite or probable) rates to those in the non-DM STEMI group (aHR, 1.394; 95% CI, 0.803–2.419; p = 0.237; Table 3), the overall ST rate was significantly higher in the DM group than in the non-DM group (aHR, 1.641; 95% CI, 1.050-2.564; p = 0.030, Table 3). Additionally, the repeat revascularization rate was significantly higher in the DM group than in the non-DM group (aHR, 1.310; 95% CI, 1.069–1.605; p = 0.009). In our study, reduced LVEF, cardiogenic shock, CPR on admission, peak troponin-I, NT-ProBNP, and lipid-lowering agents were common independent predictors of MACE in both the DM and non-DM groups (Supplementary Material 5). These are well-known independent predictors of MACE in patients with AMI [33,43,44].
Several previous studies have demonstrated that higher long-term mortality was observed in the NSTEMI group than in the STEMI group, regardless of the type of death [7,9,10,31,45]. Consistent with the results of previous studies [7,9,10,31,45], in all the patients in our study, the rates of MACE, all-cause death, CD, and non-CD were significantly higher in the NSTEMI group than in the STEMI group. However, the higher non-CD rate in the NSTEMI group was more evident in the DM group, while the higher CD rate in the NSTEMI group was more evident in the non-DM group. Hence, we believe that strategies for reducing the non-CD rate in patients with DM and those for reducing the CD rate in patients without DM could be beneficial for the NSTEMI group after successful PCI with new-generation DES. Although the sample size of the study population was too small to demonstrate meaningful results, more than 50 high-volume universities and community hospitals with facilities for primary PCI and on-site cardiac surgery participated in this study. Regarding the relatively higher incidence of multiple organ failure and CVA in the DM and NSTEMI groups than in the non-DM and STEMI groups (Supplementary Material 4), more well-established and regular follow-ups [9] as well as more focused and diverse secondary prevention therapies [31], including those involving lipid-lowering agents (Supplementary Materials 5 and 6), are required to reduce the occurrence of non-CD in patients with NSTEMI. Therefore, we believe that the results of this comparative study could provide interventional cardiologists with meaningful information regarding treatment strategies for patients with two different types of AMI according to the presence or absence of DM.
This study has several limitations. First, because we used registry data, there may have been some under-reported or missing data. Second, this study was based on discharge medications because we could not precisely determine the participants’ adherence or non-adherence to their antidiabetic drugs during the 2-year follow-up period. Third, although the interval from symptom onset to PCI was an important determinant of major clinical outcomes, this variable included many missing values in the registry data; therefore, we could not include this variable in our study. Fourth, although we performed multivariable- and PS-adjusted analyses to strengthen our results, variables not included in the KAMIR may have affected the study outcomes. Fifth, it is not certain that our population had 100% T2DM based only on the age at which diabetes was discovered. Occasionally, type 1 DM occurs in individuals aged over 30 years [48]. Moreover, there were some missing values concerning patient-reported history, such as the presence or absence of a history of ketoacidosis and other medical records indicating T2DM in the KAMIR data, owing to the registry-based nature of this study. These factors may be considered as critical limitations of this study. Sixth, this retrospective study was a long-term (November 2005 to June 2015) study of patients with AMI, which could have affected clinical outcomes. Finally, the 2-year follow-up period in this study was relatively short and may have been inadequate for determining long-term major clinical outcomes.
Because, references number 35 (Bundhun, P. K.; Li, N.; Chen, M. H. Adverse cardiovascular outcomes between insulin-treated and non-insulin treated diabetic patients after percutaneous coronary intervention: a systematic review and meta-analysis. Cardiovasc. Diabetol. 2015,14, 135) was newly added, the numbers of references were renumbered as follows,
Reference 35 (newly added), No. 35 → No. 36, No. 36 → No. 37, No. 37 → No. 38, No. 38→ No. 39, No. 39 → No. 40, No. 40 → No. 41, No. 41 → No. 42, No. 42 → No. 43, No. No. 43 → No. 44, No. 44 → No. 45, No. 45 → No. 46, No. 46 → No. 47.
- Since this study includes patients enrolled from 2005 in the registry, many diabetic patients didn’t experimented the benefits brought from the newer therapies (SGLT2i, GLP1a, etc). This point is well expressed in the limitations, but I think could be a good point to cite in the discussion section where authors talk about strategies to reduce non CV death in NSTEMI and DM patients.
Answer: We sincerely thank you for reviewer’s valuable comments and recommendations. According to reviewer’s recommendations, we added above content in the “Discussion” section as follows and we deleted above content in the “Limitation” section. Unfortunately, although the newer therapies (SGLT2i, GLP1a, etc) showed beneficial effect on reducing mortality, the type of death was not clearly defined in the recent studies.
Before,
- Discussion
The main findings of this study were as follows: (1) in the DM group, although MACE, all-cause death, CD, recurrent MI, any repeat revascularization, and ST rates were not significantly different between the NSTEMI and STEMI groups, the non-CD rate was significantly higher in the NSTEMI group than in the STEMI group; (2) in the non-DM group, the MACE, all-cause death, and CD rates were significantly higher in the NSTEMI group than in the STEMI group; (3) in the NSTEMI group, the MACE, all-cause death, non-CD, and ST rates were significantly higher in the DM group than in the non-DM group; (4) in the STEMI group, the MACE, all-cause death, CD, recurrent MI, and any repeat revascularization rates were significantly higher in the DM group than in the non-DM group; and (5) reduced LVEF (<40%), cardiogenic shock, CPR on admission, peak troponin-I level, NT-ProBNP level, and lipid-lowering agents were common independent predictors for MACE in both the DM and non-DM groups.
STEMI is the result of acute occlusion of the IRA and is associated with transmural ischemia, whereas NSTEMI is caused by transient or incomplete coronary artery occlusion, resulting in non-transmural subendocardial ischemia [4]. In previous studies [9,31], the 6-month post-discharge mortality (6.2% vs. 4.8%, respectively) and 1-year mortality (11.6% vs. 9.0%, respectively) rates were higher in the NSTEMI group than in the STEMI group. Although these two randomized studies [9,31] are valuable for estimating comparative clinical outcomes between NSTEMI and STEMI groups, they [9,31] were conducted before the new-generation DES era and were not limited to patients with DM. Hyperglycemia contributes to increased mortality and morbidity rates through an oxidative-linked mechanism [32] and may exert significant hemodynamic effects even in normal study participants [12]. Furthermore, hyperglycemia caused by oxidative stress, inflammation, apoptosis, endothelial dysfunction, hypercoagulation, and platelet aggregation [33] could damage the ischemic myocardium in patients with NSTEMI [12]. In a study by Hao et al. [12], among 890 patients with NSTEMI who underwent PCI, admission hyperglycemia was an independent predictor of a 30-day (aHR, 1.014; p < 0.001 and aHR, 1.018, p < 0.001, respectively) and 3-year MACE (aHR, 1.009; p < 0.001 and aHR, 1.017, p < 0.001, respectively) in patients with and without DM. Recently, Li et al. [13] demonstrated that the incidences of all-cause death (1.1%) and MACE (3.4%) were significantly lower in patients without a history of DM and an HbA1c level of <6.5% at admission. However, DM patients with poor glycemic control at admission experienced high rates of all-cause death (18.8%) and MACE (25%) in 350 consecutive patients with STEMI during a 2-year follow-up period. Similarly, in our study, both in the NSTEMI and STEMI groups, the rates of all-cause death (aHR, 1.755; p = 0.001 and aHR; 2.040, p < 0.001, respectively) and MACE (aHR, 1.328; p = 0.006 and aHR, 1.527; p < 0.001, respectively) were significantly higher in the DM group than in the non-DM group (Table 3). However, the study population in these studies [12,13] was limited to patients with NSTEMI or STEMI. DESs have been developed to overcome the limitations of BMS deployment, such as neointimal hyperplasia and repeat revascularization [22,24]. In the era of DES, second-generation DES (2G-DES) is the most commonly used DES because it can solve the problems of 1G-DES, such as inflammation and restenosis, and decrease the mortality rate (aHR, 1.534; p = 0.009) [14]. Hence, to provide more meaningful results and compensate for the shortcomings of the previous studies [9,12,13,31], we compared the 2-year clinical outcomes between the NSTEMI and STEMI groups according to the presence or absence of DM. The study population was strictly confined to patients with AMI who underwent successful implantation of a new-generation DES to reflect the current PCI trend. Additionally, to evaluate the long-term outcomes of the NSTEMI and STEMI groups, we excluded patients who died in the hospital (Figure 1).
Patients with NSTEMI tend to be older and have a lower rate of acute revascularization than those with STEMI [34]. In our study, in both the DM and non-DM groups, the patients in the NSTEMI group had a higher mean age than those in the STEMI group (65.5 ± 11.3 years vs. 63.0 ± 11.9, p < 0.001 and 63.5 ± 12.6 years vs. 61.4 ± 13.0, p < 0.001, respectively) and the total number of patients who underwent PCI within 24 h was lower in the NSTEMI group than that in the STEMI group (85.0% vs. 96.5%, p < 0.001 and 87.5% vs. 96.7%, p < 0.001, respectively, Table 1). Although this study did not include in-hospital outcomes, it could be stated that STEMI is a higher-risk disease in the acute phase [6,7], for which we have gathered treatment knowledge, while NSTEMI is a higher-risk disease in the chronic phase owing to the higher complexity of these patients [9,10]. Moreover, in our study, although HbA1c levels were higher in patients with STEMI, the number of patients undergoing insulin treatment was higher in the NSTEMI group. Patients with NSTEMI may have had greater decompensated diabetes at the time of MI. Insulin-treated patients with DM were associated with significantly higher short- and long-term adverse cardiovascular outcomes after PCI than those not treated with insulin therapy [35]. However, long-term comparative results between patients with insulin-treated DM and non-insulin-treated DM after NSTEMI and STEMI are very limited. Further studies are required to evaluate long-term clinical outcomes in the NSTEMI and STEMI groups. Okura et al. reported that the non-CD rate (15.1% vs. 8.4 %, p < 0.001) was significantly higher in the NSTEMI group than in the STEMI group during a median 4.3-year follow-up period [34]. In a recent study involving patients with chronic kidney disease [36], the non-CD rate (aHR, 1.960; p = 0.004) was significantly higher in the NSTEMI group than in the STEMI group. In our study, after both multivariable- (aHR, 2.223; p = 0.006) and PS-adjusted (aHR, 2.484; p = 0.004) analyses, the non-CD rate was significantly higher in the DM group than that in the STEMI group (Table 2). Furthermore, after multivariable-adjusted analysis, the non-CD rate in the NSTEMI group was significantly higher in the DM group than that in the non-DM group (aHR, 2.810; p < 0.001, Table 3). The main causes of non-CD in the DM group were multiple organ failure (0.7% vs. 0.2%, p = 0.008) and CVA (0.9% vs. 0.4%, p = 0.034) (Supplementary Material 4). In a previous report [37], CVA was independently predicted by DM (HR, 1.43; 95% CI, 1.04-1.97; p = 0.03) and was associated with a significantly increased 3-year mortality rate (aHR, 2.39; p = 0.004). DM accelerates atherosclerosis in multiple vascular beds and causes severe coronary atherosclerosis. Coronary and cerebrovascular diseases frequently coexist because of their similar pathogeneses [38]. In our study, in the non-DM group, the mortality rate (all-cause and CD) was significantly higher in the NSTEMI group than in the STEMI group; however, in the DM group, the mortality rate did not significantly differ between the two MI groups (Table 2). This result may be related to the relatively higher CD rate in the DM and STEMI groups than that in the non-DM and STEMI groups (aHR, 2.534; p < 0.001, Table 3). Additionally, this result may be related to the insignificantly different non-CD rates in the DM and STEMI groups compared to those in the non-DM and STEMI groups (aHR, 1.309; p = 0.376, Table 3).
The incidence of recurrent ischemic events after AMI is higher in the first year, and, in subsequent years, it is based on several cardiovascular risk factors [39]. Recently, Kim et al. [40] reported that in patients with AMI, the cumulative incidence of re-MI was significantly higher in the DM group than in the normoglycemia group (aHR, 1.752; 95% CI, 1.087–2.823; p = 0.021). Among all the patients in our study, the re-MI rate was significantly higher in the DM group than in the non-DM group (aHR, 1.479; 95% CI, 1.131–1.934; p = 0.004, Table 3). The re-MI rate increased continuously in the DM group, regardless of AMI type (Figure 2). In a Danish study [41], the risk of ST (definite, probable, or possible) did not differ significantly between DM and non-DM groups in patients with STEMI (aHR, 1.50; 95% CI, 0.92–2.45). Recently, a subgroup analysis of the ultrathin strut biodegradable polymer sirolimus-eluting stent versus the durable polymer everolimus-eluting stent for percutaneous coronary revascularization trial [42] showed that the 5-year ST (definite or probable) was significantly higher in the DM group than in the non-DM group (rate ratio [RR], 2.05; 95% CI, 1.45–2.90; p < 0.001) in all patients after new-generation DES implantation. Furthermore, in their study, the target lesion failure (rate ratio, 1.87; p < 0.001) and target vessel failure (rate ratio, 1.76; p < 0.001) rates were significantly higher in the DM group than in the non-DM group [42]. In our study, although patients in the DM with STEMI group showed comparable ST (definite or probable) rates to those in the non-DM STEMI group (aHR, 1.394; 95% CI, 0.803–2.419; p = 0.237; Table 3), the overall ST rate was significantly higher in the DM group than in the non-DM group (aHR, 1.641; 95% CI, 1.050-2.564; p = 0.030, Table 3). Additionally, the repeat revascularization rate was significantly higher in the DM group than in the non-DM group (aHR, 1.310; 95% CI, 1.069–1.605; p = 0.009). In our study, reduced LVEF, cardiogenic shock, CPR on admission, peak troponin-I, NT-ProBNP, and lipid-lowering agents were common independent predictors of MACE in both the DM and non-DM groups (Supplementary Material 5). These are well-known independent predictors of MACE in patients with AMI [33,43,44].
Several previous studies have demonstrated that higher long-term mortality was observed in the NSTEMI group than in the STEMI group, regardless of the type of death [7,9,10,31,45]. Consistent with the results of previous studies [7,9,10,31,45], in all the patients in our study, the rates of MACE, all-cause death, CD, and non-CD were significantly higher in the NSTEMI group than in the STEMI group. However, the higher non-CD rate in the NSTEMI group was more evident in the DM group, while the higher CD rate in the NSTEMI group was more evident in the non-DM group. Hence, we believe that strategies for reducing the non-CD rate in patients with DM and those for reducing the CD rate in patients without DM could be beneficial for the NSTEMI group after successful PCI with new-generation DES. Although the sample size of the study population was too small to demonstrate meaningful results, more than 50 high-volume universities and community hospitals with facilities for primary PCI and on-site cardiac surgery participated in this study. Regarding the relatively higher incidence of multiple organ failure and CVA in the DM and NSTEMI groups than in the non-DM and STEMI groups (Supplementary Material 4), more well-established and regular follow-ups [9] as well as more focused and diverse secondary prevention therapies [31], including those involving lipid-lowering agents (Supplementary Materials 5 and 6), are required to reduce the occurrence of non-CD in patients with NSTEMI. Therefore, we believe that the results of this comparative study could provide interventional cardiologists with meaningful information regarding treatment strategies for patients with two different types of AMI according to the presence or absence of DM.
This study has other limitations. First, because we used registry data, there may have been some under-reported and/or missing data. Second, this study was based on discharge medications since we could not precisely determine the participants’ adherence or non-adherence to their antidiabetic drugs during the 2-year follow-up period. Third, although the interval from symptom onset to PCI is an important determinant of major clinical outcomes, this variable included many missing values in the registry data. Therefore, we could not include this variable in this study. Fourth, although we performed multivariable-adjusted and PS-adjusted analyses to strengthen our results, variables not included in the KAMIR may have affected the study outcomes. Fifth, it is not certain that our population had 100% T2DM, just based on the age of discovery of their diabetes. Occasionally, T1DM can occur in individuals aged over 30 years [45]. Moreover, there were some missing values concerning patient-reported history, such as the presence or absence of a history of ketoacidosis and other medical records indicating T2DM in the KAMIR data due to the registry nature of this study. These factors may be important limitations of this study. Sixth, the progression of atherosclerosis and the development of cardiovascular events can be altered by various recently developed antidiabetic agents (e.g., sodium-glucose cotransporter 2 inhibitor, glucagon-like Peptide-1 [GLP-1] agonist) [46]. Unfortunately, we could not obtain information about the various recently developed antidiabetic agents that had been prescribed from the KAMIR registry. This is another important limitation of this study. Seventh, this retrospective study was a long-term (November 2005 to June 2015) study of patients with AMI, which could have affected the clinical outcomes. Finally, the 2-year follow-up period of this study was relatively short and may have been inadequate for determining long-term major clinical outcomes.
After,
- Discussion
The main findings of this study were as follows: (1) in the DM group, although MACE, all-cause death, CD, recurrent MI, any repeat revascularization, and ST rates were not significantly different between the NSTEMI and STEMI groups, the non-CD rate was significantly higher in the NSTEMI group than in the STEMI group; (2) in the non-DM group, the MACE, all-cause death, and CD rates were significantly higher in the NSTEMI group than in the STEMI group; (3) in the NSTEMI group, the MACE, all-cause death, non-CD, and ST rates were significantly higher in the DM group than in the non-DM group; (4) in the STEMI group, the MACE, all-cause death, CD, recurrent MI, and any repeat revascularization rates were significantly higher in the DM group than in the non-DM group; and (5) reduced LVEF (<40%), cardiogenic shock, CPR on admission, peak troponin-I level, NT-ProBNP level, and lipid-lowering agents were common independent predictors for MACE in both the DM and non-DM groups.
STEMI is the result of acute occlusion of the IRA and is associated with transmural ischemia, whereas NSTEMI is caused by transient or incomplete coronary artery occlusion, resulting in non-transmural subendocardial ischemia [4]. In previous studies [9,31], the 6-month post-discharge mortality (6.2% vs. 4.8%, respectively) and 1-year mortality (11.6% vs. 9.0%, respectively) rates were higher in the NSTEMI group than in the STEMI group. Although these two randomized studies [9,31] are valuable for estimating comparative clinical outcomes between NSTEMI and STEMI groups, they [9,31] were conducted before the new-generation DES era and were not limited to patients with DM. Hyperglycemia contributes to increased mortality and morbidity rates through an oxidative-linked mechanism [32] and may exert significant hemodynamic effects even in normal study participants [12]. Furthermore, hyperglycemia caused by oxidative stress, inflammation, apoptosis, endothelial dysfunction, hypercoagulation, and platelet aggregation [33] could damage the ischemic myocardium in patients with NSTEMI [12]. In a study by Hao et al. [12], among 890 patients with NSTEMI who underwent PCI, admission hyperglycemia was an independent predictor of a 30-day (aHR, 1.014; p < 0.001 and aHR, 1.018, p < 0.001, respectively) and 3-year MACE (aHR, 1.009; p < 0.001 and aHR, 1.017, p < 0.001, respectively) in patients with and without DM. Recently, Li et al. [13] demonstrated that the incidences of all-cause death (1.1%) and MACE (3.4%) were significantly lower in patients without a history of DM and an HbA1c level of <6.5% at admission. However, DM patients with poor glycemic control at admission experienced high rates of all-cause death (18.8%) and MACE (25%) in 350 consecutive patients with STEMI during a 2-year follow-up period. Similarly, in our study, both in the NSTEMI and STEMI groups, the rates of all-cause death (aHR, 1.755; p = 0.001 and aHR; 2.040, p < 0.001, respectively) and MACE (aHR, 1.328; p = 0.006 and aHR, 1.527; p < 0.001, respectively) were significantly higher in the DM group than in the non-DM group (Table 3). However, the study population in these studies [12,13] was limited to patients with NSTEMI or STEMI. DESs have been developed to overcome the limitations of BMS deployment, such as neointimal hyperplasia and repeat revascularization [22,24]. In the era of DES, second-generation DES (2G-DES) is the most commonly used DES because it can solve the problems of 1G-DES, such as inflammation and restenosis, and decrease the mortality rate (aHR, 1.534; p = 0.009) [14]. Hence, to provide more meaningful results and compensate for the shortcomings of the previous studies [9,12,13,31], we compared the 2-year clinical outcomes between the NSTEMI and STEMI groups according to the presence or absence of DM. The study population was strictly confined to patients with AMI who underwent successful implantation of a new-generation DES to reflect the current PCI trend. Additionally, to evaluate the long-term outcomes of the NSTEMI and STEMI groups, we excluded patients who died in the hospital (Figure 1).
Patients with NSTEMI tend to be older and have a lower rate of acute revascularization than those with STEMI [34]. In our study, in both the DM and non-DM groups, the patients in the NSTEMI group had a higher mean age than those in the STEMI group (65.5 ± 11.3 years vs. 63.0 ± 11.9, p < 0.001 and 63.5 ± 12.6 years vs. 61.4 ± 13.0, p < 0.001, respectively) and the total number of patients who underwent PCI within 24 h was lower in the NSTEMI group than that in the STEMI group (85.0% vs. 96.5%, p < 0.001 and 87.5% vs. 96.7%, p < 0.001, respectively, Table 1). Although this study did not include in-hospital outcomes, it could be stated that STEMI is a higher-risk disease in the acute phase [6,7], for which we have gathered treatment knowledge, while NSTEMI is a higher-risk disease in the chronic phase owing to the higher complexity of these patients [9,10]. Moreover, in our study, although HbA1c levels were higher in patients with STEMI, the number of patients undergoing insulin treatment was higher in the NSTEMI group. Patients with NSTEMI may have had greater decompensated diabetes at the time of MI. Insulin-treated patients with DM were associated with significantly higher short- and long-term adverse cardiovascular outcomes after PCI than those not treated with insulin therapy [35]. However, long-term comparative results between patients with insulin-treated DM and non-insulin-treated DM after NSTEMI and STEMI are very limited. Further studies are required to evaluate long-term clinical outcomes in the NSTEMI and STEMI groups. Okura et al. reported that the non-CD rate (15.1% vs. 8.4 %, p < 0.001) was significantly higher in the NSTEMI group than in the STEMI group during a median 4.3-year follow-up period [34]. In a recent study involving patients with chronic kidney disease [36], the non-CD rate (aHR, 1.960; p = 0.004) was significantly higher in the NSTEMI group than in the STEMI group. In our study, after both multivariable- (aHR, 2.223; p = 0.006) and PS-adjusted (aHR, 2.484; p = 0.004) analyses, the non-CD rate was significantly higher in the DM group than that in the STEMI group (Table 2). Furthermore, after multivariable-adjusted analysis, the non-CD rate in the NSTEMI group was significantly higher in the DM group than that in the non-DM group (aHR, 2.810; p < 0.001, Table 3). The main causes of non-CD in the DM group were multiple organ failure (0.7% vs. 0.2%, p = 0.008) and CVA (0.9% vs. 0.4%, p = 0.034) (Supplementary Material 4). In a previous report [37], CVA was independently predicted by DM (HR, 1.43; 95% CI, 1.04-1.97; p = 0.03) and was associated with a significantly increased 3-year mortality rate (aHR, 2.39; p = 0.004). DM accelerates atherosclerosis in multiple vascular beds and causes severe coronary atherosclerosis. Coronary and cerebrovascular diseases frequently coexist because of their similar pathogeneses [38]. In our study, in the non-DM group, the mortality rate (all-cause and CD) was significantly higher in the NSTEMI group than in the STEMI group; however, in the DM group, the mortality rate did not significantly differ between the two MI groups (Table 2). This result may be related to the relatively higher CD rate in the DM and STEMI groups than that in the non-DM and STEMI groups (aHR, 2.534; p < 0.001, Table 3). Additionally, this result may be related to the insignificantly different non-CD rates in the DM and STEMI groups compared to those in the non-DM and STEMI groups (aHR, 1.309; p = 0.376, Table 3).
The incidence of recurrent ischemic events after AMI is higher in the first year, and, in subsequent years, it is based on several cardiovascular risk factors [39]. Recently, Kim et al. [40] reported that in patients with AMI, the cumulative incidence of re-MI was significantly higher in the DM group than in the normoglycemia group (aHR, 1.752; 95% CI, 1.087–2.823; p = 0.021). Among all the patients in our study, the re-MI rate was significantly higher in the DM group than in the non-DM group (aHR, 1.479; 95% CI, 1.131–1.934; p = 0.004, Table 3). The re-MI rate increased continuously in the DM group, regardless of AMI type (Figure 2). In a Danish study [41], the risk of ST (definite, probable, or possible) did not differ significantly between DM and non-DM groups in patients with STEMI (aHR, 1.50; 95% CI, 0.92–2.45). Recently, a subgroup analysis of the ultrathin strut biodegradable polymer sirolimus-eluting stent versus the durable polymer everolimus-eluting stent for percutaneous coronary revascularization trial [42] showed that the 5-year ST (definite or probable) was significantly higher in the DM group than in the non-DM group (rate ratio [RR], 2.05; 95% CI, 1.45–2.90; p < 0.001) in all patients after new-generation DES implantation. Furthermore, in their study, the target lesion failure (rate ratio, 1.87; p < 0.001) and target vessel failure (rate ratio, 1.76; p < 0.001) rates were significantly higher in the DM group than in the non-DM group [42]. In our study, although patients in the DM with STEMI group showed comparable ST (definite or probable) rates to those in the non-DM STEMI group (aHR, 1.394; 95% CI, 0.803–2.419; p = 0.237; Table 3), the overall ST rate was significantly higher in the DM group than in the non-DM group (aHR, 1.641; 95% CI, 1.050-2.564; p = 0.030, Table 3). Additionally, the repeat revascularization rate was significantly higher in the DM group than in the non-DM group (aHR, 1.310; 95% CI, 1.069–1.605; p = 0.009). In our study, reduced LVEF, cardiogenic shock, CPR on admission, peak troponin-I, NT-ProBNP, and lipid-lowering agents were common independent predictors of MACE in both the DM and non-DM groups (Supplementary Material 5). These are well-known independent predictors of MACE in patients with AMI [33,43,44].
Several previous studies have demonstrated that higher long-term mortality was observed in the NSTEMI group than in the STEMI group, regardless of the type of death [7,9,10,31,45]. Consistent with the results of previous studies [7,9,10,31,45], in all the patients in our study, the rates of MACE, all-cause death, CD, and non-CD were significantly higher in the NSTEMI group than in the STEMI group. However, the higher non-CD rate in the NSTEMI group was more evident in the DM group, while the higher CD rate in the NSTEMI group was more evident in the non-DM group. Hence, we believe that strategies for reducing the non-CD rate in patients with DM and those for reducing the CD rate in patients without DM could be beneficial for the NSTEMI group after successful PCI with new-generation DES. Although the sample size of the study population was too small to demonstrate meaningful results, more than 50 high-volume universities and community hospitals with facilities for primary PCI and on-site cardiac surgery participated in this study. Regarding the relatively higher incidence of multiple organ failure and CVA in the DM and NSTEMI groups than in the non-DM and STEMI groups (Supplementary Material 4), more well-established and regular follow-ups [9] as well as more focused and diverse secondary prevention therapies [31], including those involving lipid-lowering agents (Supplementary Materials 5 and 6), are required to reduce the occurrence of non-CD in patients with NSTEMI. Since this study enrolled patients (from the registry) from 2005, many patients with diabetes did not experience the benefits of newer therapies (such as sodium-glucose cotransporter 2 inhibitor, glucagon-like Peptide-1 receptor agonist). These medications have been shown to reduce the risk of cardiovascular events when compared with that of the controls [46,47]. Therefore, recent guidelines support the incorporation of these newer agents with cardiovascular benefits into routine clinical practice and screening of patients who are at a high risk of cardiovascular disease. Unfortunately, because we could not obtain information about the various recently developed antidiabetic agents that had been prescribed from the KAMIR registry, we could not provide information concerning the effects of these drugs on the long-term clinical outcomes in this study. Finally, we believe that the results of this comparative study could provide interventional cardiologists with meaningful information regarding treatment strategies for patients with two different types of AMI according to the presence or absence of DM.
This study has several limitations. First, because we used registry data, there may have been some under-reported or missing data. Second, this study was based on discharge medications because we could not precisely determine the participants’ adherence or non-adherence to their antidiabetic drugs during the 2-year follow-up period. Third, although the interval from symptom onset to PCI was an important determinant of major clinical outcomes, this variable included many missing values in the registry data; therefore, we could not include this variable in our study. Fourth, although we performed multivariable- and PS-adjusted analyses to strengthen our results, variables not included in the KAMIR may have affected the study outcomes. Fifth, it is not certain that our population had 100% T2DM based only on the age at which diabetes was discovered. Occasionally, type 1 DM occurs in individuals aged over 30 years [48]. Moreover, there were some missing values concerning patient-reported history, such as the presence or absence of a history of ketoacidosis and other medical records indicating T2DM in the KAMIR data, owing to the registry-based nature of this study. These factors may be considered as critical limitations of this study. Sixth, this retrospective study was a long-term (November 2005 to June 2015) study of patients with AMI, which could have affected clinical outcomes. Finally, the 2-year follow-up period in this study was relatively short and may have been inadequate for determining long-term major clinical outcomes.
Because, references number 46 (McKee, A.; Al-Khazaali, A.; Albert, S. G. Glucagon-like Peptide-1 Receptor Agonists versus Sodium-Glucose Cotransporter Inhibitors for Treatment of T2DM. J Endocr Soc. 2020, 4, bvaa037) was newly added, the numbers of references were renumbered as follows,
Reference 46 (newly added), No. 46 → No.48, No. 48 → No.47.
- From statistical methods it seems that a test for normal distribution wasn’t performed, indeed some variables had a SD higher than the mean value (I.e. NTproBNP). This results let me think for a non normal distribution of the variable. Please fix the issue, reporting median and interquartile range.
Answer: We sincerely thank you for reviewer’s valuable comments and recommendations. According to reviewer’s recommendations, we revised the Table 1 as follows using median and interquartile range.
Before,
After,
Before (Supplementary material 2),
After (Supplementary material 2),
Before (Supplementary material 3),
After (Supplementary material 3),
- Authors provided several KM curves. KM curves from A to D show a cumulative incidence scale ranging from 0 to 8, while E-G curves from 0 to 6. Please uniform the scales.
Answer: We sincerely thank you for reviewer’s valuable comments and recommendations. According to reviewer’s recommendations, we unified the cumulative incidence scale ranging from 0 to 8 in the KM curves as follows,
Before
After,
Dear respected professor reviewer 1,
We sincerely thank you for providing us an opportunity for resubmission. We also sincerely thank you for your thoughtful evaluation of our work and the valuable suggestions that helped us to improve the manuscript. We faithfully revised our paper according to your comments and recommendations as possible as.
Thank you for your consideration and we look forward to hearing from you.
Sincerely,
Seung-Woon Rha, MD., Ph.D., FACC, FAHA, FESC, FSCAI, FAPSIC.
Yong Hoon Kim, M.D., Ph.D.

Reviewer 2 Report
The authors aimed to evaluate, in a non-randomized multicenter observational retrospective cohort study, the 2-year comparative clinical outcomes between the two different types of AMI (NSTEMI versus STEMI) in DM and non-DM populations who underwent successful PCI with a new-generation DES.
The study covers some issues that have been overlooked in other similar topics. The structure of the manuscript appears adequate and well divided in the sections. Moreover, the study is easy to follow, but some issues should be improved. Some of the comments that would improve the overall quality of the study are:
a. Authors must pay attention to the technical terms acronyms they used in the text.
b. English language needs to be revised.
c. Conclusion Section: This paragraph required a general revision to eliminate redundant sentences and to add some "take-home message".
Author Response
Comments and Suggestions for Authors
The authors aimed to evaluate, in a non-randomized multicenter observational retrospective cohort study, the 2-year comparative clinical outcomes between the two different types of AMI (NSTEMI versus STEMI) in DM and non-DM populations who underwent successful PCI with a new-generation DES.
The study covers some issues that have been overlooked in other similar topics. The structure of the manuscript appears adequate and well divided in the sections. Moreover, the study is easy to follow, but some issues should be improved. Some of the comments that would improve the overall quality of the study are:
- Authors must pay attention to the technical terms acronyms they used in the text.
Answer: We sincerely thank you for reviewer’s valuable comments and recommendations. Moreover, we totally agree with reviewer’s comments. We carefully reviewed the technical terms acronyms they used in the text and we revised main manuscript as follows,
After,
- Introduction
Diabetes mellitus (DM), particularly type 2 DM (T2DM), is one of the most important threats to public health in the twenty-first century [1]. In the Harmonizing Outcomes with RevasculariZatiON and Stents in Acute Myocardial Infarction (AMI) trial [2], patients with DM and those with newly diagnosed DM had higher 3-year death rates than non-DM patients (11%, 12%, and 6%, respectively). Thrombus formation after rupture or erosion of vulnerable atherosclerotic plaques is a common pathophysiology in both non-ST-segment-elevation myocardial infarction (NSTEMI) and ST-segment-elevation myocardial infarction (STEMI) [3]. However, patients with NSTEMI have partial or intermittent occlusion of the coronary artery, whereas patients with STEMI often have complete occlusion [4]. Additionally, after a certain duration of complete ischemia, there are no interventions that can salvage the ischemic myocardium [5]. Because cardiogenic shock complicates an increasing number of STEMI cases [6], Polonski et al. suggested that in-hospital outcomes were worse in patients with STEMI than in those with NSTEMI. Patients with NSTEMI have a greater prevalence of comorbidities [7] and have received relatively fewer guideline-based treatments than patients with STEMI [8]. Hence, long-term mortality is higher in patients with NSTEMI than in those with STEMI [9,10]. Until now, comparative results between NSTEMI and STEMI have been conflicting, and further discussion is needed [10,11]. Because recent reports concerning long-term clinical outcomes in patients with and without DM are confined to patients with NSTEMI [12] or STEMI [13], data on head-to-head comparisons between long-term clinical outcomes in NSTEMI and STEMI in patients with and without DM are scarce. Currently, new-generation drug-eluting stents (DESs) have nearly replaced bare-metal stents (BMSs) and first-generation DESs (1G-DESs) for routine percutaneous coronary intervention (PCI) [14]. New-generation DESs are more effective than 1G-DESs in reducing major clinical outcomes in patients with DM [14]. Furthermore, to the best of our knowledge, no specific large-scale study has compared the long-term clinical outcomes between the NSTEMI and STEMI groups in patients with and without DM after PCI using new-generation DESs to reflect current real-world practice. Hence, this study, using a new-generation DES, evaluated the 2-year comparative clinical outcomes between two different types of AMI (NSTEMI versus STEMI) in patients with DM and non-DM who underwent successful PCI.
- Methods
2.1. Study Population
This non-randomized multicenter observational retrospective cohort study enrolled 21,343 patients with AMI aged ≥30 years at the onset of DM who underwent successful PCI with newer-generation DESs between November 2005 and June 2015; data was obtained from the Korea AMI Registry (KAMIR). To ensure that only individuals with T2DM were included, patients aged <30 years at the onset of DM were excluded based on a previous study [15]. KAMIR was established in November 2005 and involves more than 50 communities and teaching hospitals in South Korea [16]. Patients with the following conditions were excluded: incomplete laboratory results including unidentified results of blood hemoglobin (Hb) A1c and blood glucose (n = 8314; 39.0%), lost to follow-up (n = 1067; 5.0%), and in-hospital death (n = 164; 0.8%). After exclusion, 11,798 patients with AMI who underwent successful PCI using new-generation DESs were included. The patients were classified into DM (n = 5092; 43.2%) and non-DM (n = 6706; 56.8%) groups (Figure 1). Thereafter, these two groups were further sub-classified into NSTEMI (group A, n = 2399 [47.1%] and group C, n = 2694 [40.2%], respectively) and STEMI (group B, n = 2693 [52.9%] and group D, n = 4012 [59.8%], respectively). This study was conducted in accordance with the ethical guidelines of the 2004 Declaration of Helsinki and was approved by the ethics committee of each participating center and the Chonnam National University Hospital Institutional Review Board ethics committee (CNUH-2011-172). All 11,798 patients included in the study provided written informed consent prior to enrollment and completed a 2-year clinical follow-up through face-to-face interviews, phone calls, or chart reviews. An independent event adjudication committee evaluated all clinical events. The event adjudication process has been previously described by KAMIR investigators [16].
2.2. Percutaneous Coronary Intervention and Medical Treatment
Diagnostic coronary angiography (CAG) and PCI were performed according to the general guidelines [17]. Loading doses of aspirin (200–300 mg), clopidogrel (300–600 mg), ticagrelor (180 mg), and prasugrel (60 mg) were administered to all the enrolled patients prior to PCI. Subsequently, dual antiplatelet therapy (DAPT; a combination of aspirin [100 mg/day] with clopidogrel [75 mg/day], or ticagrelor [90 mg twice a day], or prasugrel [5–10 mg/day]) was recommended for >12 months. Based on previous reports [18,19], triple antiplatelet therapy was administered (TAPT;100 mg cilostazol was administered twice a day in addition to the dual antiplatelet therapy) at the discretion of the individual operator. Moreover, the access site, revascularization strategy, and DES selection were assigned to the individual operators and were to be performed at their own discretion.
2.3. Study Definitions and Clinical Outcomes
The DM group included patients with HbA1c, fasting plasma glucose, and/or random plasma glucose levels of ≥6.5%, ≥126 mg/dL (7.0 mmol/L), and ≥200 mg/dL (11.1 mmol/L) at index hospitalization, respectively, according to the American Association clinical practice recommendations [20]. In addition to their medical history, patients with known diabetes, for which they received medical treatment (insulin or antidiabetic), or newly diagnosed diabetes were also included in the DM group. NSTEMI was defined as the absence of persistent ST-segment elevation with increased levels of cardiac biomarkers and appropriate clinical context [21,22]. STEMI was defined as follows: ongoing chest pain and electrocardiogram findings on admission showing an ST-segment-elevation in at least two contiguous leads of ≥2 mm (0.2 mV) in men or ≥1.5 mm (0.15 mV) in women in leads V2–V3 and/or ≥1 mm (0.1 mV) in other contiguous chest leads, limb leads, or new-onset left bundle branch block [23,24]. The primary clinical outcome of this study was the occurrence of major adverse cardiac events (MACE), defined as all-cause death, recurrent myocardial infarction (re-MI), or repeat coronary revascularization, including target lesion revascularization (TLR), target vessel revascularization (TVR), and non-TVR. The secondary outcome was definite or probable stent thrombosis during the 2-year follow-up period. All-cause death was considered as cardiac death (CD) unless an undisputed non-cardiac cause was present [25]. The definitions of re-MI, target lesion revascularization, TVR, and non-TVR have been previously published [26]. The types of new-generation DESs used are listed in Table 1. Glomerular function was calculated using the Chronic Kidney Disease Epidemiology Collaboration equation for the estimated glomerular filtration rate (eGFR) [27]. In our study, patients who required multivessel PCI included those who underwent PCI of the non-infarct-related artery (IRA) during index PCI of the IRA or who underwent staged PCI for the non-IRA within the index hospitalization. Therefore, patients who underwent staged PCI after discharge were excluded from this study because reperfusion timing could have acted as a bias.
2.4. Statistical Analyses
For continuous variables, between-group differences were evaluated using unpaired t tests. Data are expressed as mean ± standard deviation. For discrete variables, between-group differences were expressed as counts and percentages and analyzed using the chi-squared or Fisher’s exact test. We tested all variables with a p-value of <0.001 in the univariate analysis between the NSTEMI and STEMI groups. After univariate analysis, we performed a multicollinearity test [28] between the included variables to confirm that there was no definite collinearity between them (Supplementary Material 1). The variance inflation factor (VIF) values were calculated to measure the degree of multicollinearity among the variables. A variance inflation factor of >5 indicates a high correlation [29]. Multicollinearity was considered when the tolerance value was <0.1 [30] or the condition index was >10 [29]. The variables included in the multivariable Cox regression analysis were as follows: male sex, age, left ventricular ejection fraction (LVEF), body mass index, systolic blood pressure (SBP), diastolic blood pressure (DBP), cardiogenic shock, Killip class I/II, cardiopulmonary resuscitation (CPR) on admission, hypertension, dyslipidemia, previous myocardial infarction, previous PCI, previous coronary artery bypass graft (CABG), previous heart failure (HF), previous cerebrovascular accidents (CVA), current smoker, peak creatine kinase myocardial band (CK-MB), peak troponin-I, N-terminal pro-brain natriuretic peptide (NT-ProBNP), serum creatinine, eGFR, total cholesterol, triglyceride, low-density lipoprotein cholesterol, clopidogrel, ticagrelor, prasugrel, angiotensin-converting enzyme inhibitor (ACEI), angiotensin receptor blocker (ARB), beta-blocker, calcium channel blocker, and lipid-lowering agent. Moreover, to adjust for potential confounders, a propensity score (PS)-adjusted analysis was performed using a logistic regression model. We tested all available variables that could be of potential relevance, including baseline clinical, angiographic, and procedural factors (Table 1). The c-statistic for the PS-matched analysis in this study was 0.712. Patients in the NSTEMI group were matched to those in the STEMI group (1:1) according to PSs using the nearest available pair-matching method. The patients were matched using a caliper width of 0.01. This procedure yielded 5,768 well-matched pairs (Supplementary Material 2). Various clinical outcomes were estimated using Kaplan–Meier curve analysis, and group differences were compared using the log-rank test. Statistical significance was defined as a 2-tailed p-value of <0.05. All statistical analyses were performed using the SPSS software version 20 (IBM, Armonk, NY, USA).
- Results
3.1. Baseline Characteristics
Table 1 and Supplementary Material 3 show the baseline, laboratory, angiographic, and procedural characteristics of the study population. In both the DM and non-DM groups, patients in the NSTEMI group had a higher mean age, mean values of LVEF, systolic blood pressure, diastolic blood pressure, NT-ProBNP levels, diameter of the deployed stent, length of deployed stent, and mean number of deployed stents than patients in the STEMI group. The numbers of the following patients were higher in the NSTEMI group: with hypertension, dyslipidemia, previous histories of PCI and CVA, prescribed with ARB, left main coronary artery and left circumflex artery as the IRA and treated vessels, American College of Cardiology/American Heart Association (ACC/AHA) type B2 lesions and multivessel diseases, and those who received transradial approach and underwent intravascular ultrasound examination. The STEMI group included the following patients: a higher number of men; cardiogenic shock; underwent CPR on admission; current smokers; prescribed with ACEI and glycoprotein II/IIIa; right coronary artery as an IRA and treated vessel; American College of Cardiology/American Heart Association type C lesions; single-vessel disease and pre-PCI thrombolysis in myocardial infarction (TIMI) flow grade 0/1, and those who underwent PCI within 24 h compared to those in the NSTEMI group. In both the NSTEMI and STEMI groups, patients in the DM group had a higher mean age, mean body mass index values, NT-ProBNP levels, triglyceride levels, and mean number of deployed stents than those in the non-DM group (Supplementary Table S3). The number of patients with hypertension, dyslipidemia, previous history of MI, PCI, coronary artery bypass graft, and CVA, those prescribed ARB, and those with RCA as a treated vessel and multivessel disease were also higher in the DM group. The non-DM group included the following patients: a higher number of men; underwent CPR on admission; current smokers; those prescribed with ticagrelor, ACEI, lipid-lowering agent, and glycoprotein II/IIIa; and those with single-vessel disease compared to those in the DM group. The mean values of LVEF, peak CK-MB, total cholesterol, low-density lipoprotein cholesterol, and number of patients with pre-PCI TIMI flow grade 0/1 were also higher in the non-DM group.
3.2. Clinical Outcomes
The cumulative incidences of major clinical outcomes during the 2-year follow-up period are summarized in Tables 2 and 3 and Fig. 2. In the DM group, after multivariable-adjusted analysis, the incidence rates of MACE (adjusted hazard ratio [aHR], 1.101; 95% confidence interval [CI], 0.900–1.349; p = 0.349), all-cause death, CD, re-MI, any repeat revascularization, and ST were not significantly different between the NSTEMI and STEMI groups (Table 2). However, the non-CD rate was significantly higher in the NSTEMI group than that in the STEMI group (aHR, 2.223; 95% CI, 1.257–3.931; p = 0.006). After PS-adjusted analysis, the non-CD rate was significantly higher in the NSTEMI group than in the STEMI group (aHR, 2.484; 95% CI, 1.326–4.651; p = 0.004). In the non-DM group, after multivariable-adjusted analysis, the rates of MACE (aHR, 1.393; p = 0.002), all-cause death (aHR, 2.089; p < 0.01), and CD (aHR, 2.799; 95% CI, p < 0.001) were significantly higher in the NSTEMI group than in the STEMI group (Table 2). After PS-adjusted analysis, the rates of MACE (aHR, 1.543; p < 0.001), all-cause death (aHR, 2.172; p < 0.001), and CD (aHR, 2.882; p < 0.001) were also higher in the NSTEMI group than in the STEMI group. In all patients, after both multivariable- and PS-adjusted analyses, the rates of MACE (aHR, 1.240; p = 0.005 and aHR, 1.298; p = 0.002, respectively), all-cause death (aHR, 1.561; p < 0.001 and aHR, 1.653; p < 0.001, respectively), CD (aHR, 1.389; p = 0.039 and aHR, 1.499; p = 0.022, respectively), and non-CD rate (aHR, 1.834; p = 0.004 and aHR, 1.977; p = 0.004, respectively) were significantly higher in the NSTEMI group than in the STEMI group. Table 3 compares the clinical outcomes of the DM and non-DM groups. In the NSTEMI group, after multivariable-adjusted analysis, the rates of MACE (aHR, 1.328; p = 0.006), all-cause death (aHR, 1.755; p = 0.001), non-CD rate (aHR, 2.810; p < 0.001), and ST (aHR, 2.287; p = 0.044) were significantly higher in the DM group than in the non-DM group. However, the CD rates were similar between the two groups. In the STEMI group, the rates of MACE (aHR, 1.527; p < 0.001), all-cause death (aHR, 2.040; p < 0.001), CD (aHR, 2.534; p < 0.001), re-MI (aHR, 1.509; p = 0.029), and any repeat revascularization (aHR, 1.376; p = 0.024) were significantly higher in the DM group than in the non-DM group. However, the non-CD rates were similar between the two groups. Overall, all clinical outcomes were worse in the DM group than those in the non-DM group. Supplementary Material 4 shows the causes of non-CD in this study. In the DM group, the multiple organ failure (0.7% vs. 0.2%, p = 0.008) and CVA (0.9% vs. 0.4%, p = 0.034) rates were significantly higher in the NSTEMI group than in the STEMI group. However, in the non-DM group, none of the causes of non-CD, including multiple organ failure and CVA, were significantly different between the NSTEMI and STEMI groups. Supplementary Material 5 shows the independent predictors of MACE. Reduced LVEF (<40%), cardiogenic shock, CPR on admission, peak troponin-I, NT-ProBNP, and lipid-lowering agents were common independent predictors of MACE in both the DM and non-DM groups. Supplementary Material 6 shows the subgroup analysis for MACE in patients with and without DM. In the DM group, patients without cardiogenic shock, hypertension, or dyslipidemia and patients who received lipid-lowering agents or a deployed stent with a mean diameter of ≥3 mm had a lower MACE rate in the STEMI group than in the NSTEMI group. In the non-DM group, patients without dyslipidemia and those who received lipid-lowering agents had a lower MACE rate in the STEMI group than in the NSTEMI group.
- Discussion
The main findings of this study were as follows: (1) in the DM group, although MACE, all-cause death, CD, recurrent MI, any repeat revascularization, and ST rates were not significantly different between the NSTEMI and STEMI groups, the non-CD rate was significantly higher in the NSTEMI group than in the STEMI group; (2) in the non-DM group, the MACE, all-cause death, and CD rates were significantly higher in the NSTEMI group than in the STEMI group; (3) in the NSTEMI group, the MACE, all-cause death, non-CD, and ST rates were significantly higher in the DM group than in the non-DM group; (4) in the STEMI group, the MACE, all-cause death, CD, recurrent MI, and any repeat revascularization rates were significantly higher in the DM group than in the non-DM group; and (5) reduced LVEF (<40%), cardiogenic shock, CPR on admission, peak troponin-I level, NT-ProBNP level, and lipid-lowering agents were common independent predictors for MACE in both the DM and non-DM groups.
STEMI is the result of acute occlusion of the IRA and is associated with transmural ischemia, whereas NSTEMI is caused by transient or incomplete coronary artery occlusion, resulting in non-transmural subendocardial ischemia [4]. In previous studies [9,31], the 6-month post-discharge mortality (6.2% vs. 4.8%, respectively) and 1-year mortality (11.6% vs. 9.0%, respectively) rates were higher in the NSTEMI group than in the STEMI group. Although these two randomized studies [9,31] are valuable for estimating comparative clinical outcomes between NSTEMI and STEMI groups, they [9,31] were conducted before the new-generation DES era and were not limited to patients with DM. Hyperglycemia contributes to increased mortality and morbidity rates through an oxidative-linked mechanism [32] and may exert significant hemodynamic effects even in normal study participants [12]. Furthermore, hyperglycemia caused by oxidative stress, inflammation, apoptosis, endothelial dysfunction, hypercoagulation, and platelet aggregation [33] could damage the ischemic myocardium in patients with NSTEMI [12]. In a study by Hao et al. [12], among 890 patients with NSTEMI who underwent PCI, admission hyperglycemia was an independent predictor of a 30-day (aHR, 1.014; p < 0.001 and aHR, 1.018, p < 0.001, respectively) and 3-year MACE (aHR, 1.009; p < 0.001 and aHR, 1.017, p < 0.001, respectively) in patients with and without DM. Recently, Li et al. [13] demonstrated that the incidences of all-cause death (1.1%) and MACE (3.4%) were significantly lower in patients without a history of DM and an HbA1c level of <6.5% at admission. However, DM patients with poor glycemic control at admission experienced high rates of all-cause death (18.8%) and MACE (25%) in 350 consecutive patients with STEMI during a 2-year follow-up period. Similarly, in our study, both in the NSTEMI and STEMI groups, the rates of all-cause death (aHR, 1.755; p = 0.001 and aHR; 2.040, p < 0.001, respectively) and MACE (aHR, 1.328; p = 0.006 and aHR, 1.527; p < 0.001, respectively) were significantly higher in the DM group than in the non-DM group (Table 3). However, the study population in these studies [12,13] was limited to patients with NSTEMI or STEMI. DESs have been developed to overcome the limitations of BMS deployment, such as neointimal hyperplasia and repeat revascularization [22,24]. In the era of DES, second-generation DES (2G-DES) is the most commonly used DES because it can solve the problems of 1G-DES, such as inflammation and restenosis, and decrease the mortality rate (aHR, 1.534; p = 0.009) [14]. Hence, to provide more meaningful results and compensate for the shortcomings of the previous studies [9,12,13,31], we compared the 2-year clinical outcomes between the NSTEMI and STEMI groups according to the presence or absence of DM. The study population was strictly confined to patients with AMI who underwent successful implantation of a new-generation DES to reflect the current PCI trend. Additionally, to evaluate the long-term outcomes of the NSTEMI and STEMI groups, we excluded patients who died in the hospital (Figure 1).
Patients with NSTEMI tend to be older and have a lower rate of acute revascularization than those with STEMI [34]. In our study, in both the DM and non-DM groups, the patients in the NSTEMI group had a higher mean age than those in the STEMI group (65.5 ± 11.3 years vs. 63.0 ± 11.9, p < 0.001 and 63.5 ± 12.6 years vs. 61.4 ± 13.0, p < 0.001, respectively) and the total number of patients who underwent PCI within 24 h was lower in the NSTEMI group than that in the STEMI group (85.0% vs. 96.5%, p < 0.001 and 87.5% vs. 96.7%, p < 0.001, respectively, Table 1). Although this study did not include in-hospital outcomes, it could be stated that STEMI is a higher-risk disease in the acute phase [6,7], for which we have gathered treatment knowledge, while NSTEMI is a higher-risk disease in the chronic phase owing to the higher complexity of these patients [9,10]. Moreover, in our study, although HbA1c levels were higher in patients with STEMI, the number of patients undergoing insulin treatment was higher in the NSTEMI group. Patients with NSTEMI may have had greater decompensated diabetes at the time of MI. Insulin-treated patients with DM were associated with significantly higher short- and long-term adverse cardiovascular outcomes after PCI than those not treated with insulin therapy [35]. However, long-term comparative results between patients with insulin-treated DM and non-insulin-treated DM after NSTEMI and STEMI are very limited. Further studies are required to evaluate long-term clinical outcomes in the NSTEMI and STEMI groups. Okura et al. reported that the non-CD rate (15.1% vs. 8.4 %, p < 0.001) was significantly higher in the NSTEMI group than in the STEMI group during a median 4.3-year follow-up period [34]. In a recent study involving patients with chronic kidney disease [36], the non-CD rate (aHR, 1.960; p = 0.004) was significantly higher in the NSTEMI group than in the STEMI group. In our study, after both multivariable- (aHR, 2.223; p = 0.006) and PS-adjusted (aHR, 2.484; p = 0.004) analyses, the non-CD rate was significantly higher in the DM group than that in the STEMI group (Table 2). Furthermore, after multivariable-adjusted analysis, the non-CD rate in the NSTEMI group was significantly higher in the DM group than that in the non-DM group (aHR, 2.810; p < 0.001, Table 3). The main causes of non-CD in the DM group were multiple organ failure (0.7% vs. 0.2%, p = 0.008) and CVA (0.9% vs. 0.4%, p = 0.034) (Supplementary Material 4). In a previous report [37], CVA was independently predicted by DM (HR, 1.43; 95% CI, 1.04-1.97; p = 0.03) and was associated with a significantly increased 3-year mortality rate (aHR, 2.39; p = 0.004). DM accelerates atherosclerosis in multiple vascular beds and causes severe coronary atherosclerosis. Coronary and cerebrovascular diseases frequently coexist because of their similar pathogeneses [38]. In our study, in the non-DM group, the mortality rate (all-cause and CD) was significantly higher in the NSTEMI group than in the STEMI group; however, in the DM group, the mortality rate did not significantly differ between the two MI groups (Table 2). This result may be related to the relatively higher CD rate in the DM and STEMI groups than that in the non-DM and STEMI groups (aHR, 2.534; p < 0.001, Table 3). Additionally, this result may be related to the insignificantly different non-CD rates in the DM and STEMI groups compared to those in the non-DM and STEMI groups (aHR, 1.309; p = 0.376, Table 3).
The incidence of recurrent ischemic events after AMI is higher in the first year, and, in subsequent years, it is based on several cardiovascular risk factors [39]. Recently, Kim et al. [40] reported that in patients with AMI, the cumulative incidence of re-MI was significantly higher in the DM group than in the normoglycemia group (aHR, 1.752; 95% CI, 1.087–2.823; p = 0.021). Among all the patients in our study, the re-MI rate was significantly higher in the DM group than in the non-DM group (aHR, 1.479; 95% CI, 1.131–1.934; p = 0.004, Table 3). The re-MI rate increased continuously in the DM group, regardless of AMI type (Figure 2). In a Danish study [41], the risk of ST (definite, probable, or possible) did not differ significantly between DM and non-DM groups in patients with STEMI (aHR, 1.50; 95% CI, 0.92–2.45). Recently, a subgroup analysis of the ultrathin strut biodegradable polymer sirolimus-eluting stent versus the durable polymer everolimus-eluting stent for percutaneous coronary revascularization trial [42] showed that the 5-year ST (definite or probable) was significantly higher in the DM group than in the non-DM group (rate ratio [RR], 2.05; 95% CI, 1.45–2.90; p < 0.001) in all patients after new-generation DES implantation. Furthermore, in their study, the target lesion failure (rate ratio, 1.87; p < 0.001) and target vessel failure (rate ratio, 1.76; p < 0.001) rates were significantly higher in the DM group than in the non-DM group [42]. In our study, although patients in the DM with STEMI group showed comparable ST (definite or probable) rates to those in the non-DM STEMI group (aHR, 1.394; 95% CI, 0.803–2.419; p = 0.237; Table 3), the overall ST rate was significantly higher in the DM group than in the non-DM group (aHR, 1.641; 95% CI, 1.050-2.564; p = 0.030, Table 3). Additionally, the repeat revascularization rate was significantly higher in the DM group than in the non-DM group (aHR, 1.310; 95% CI, 1.069–1.605; p = 0.009). In our study, reduced LVEF, cardiogenic shock, CPR on admission, peak troponin-I, NT-ProBNP, and lipid-lowering agents were common independent predictors of MACE in both the DM and non-DM groups (Supplementary Material 5). These are well-known independent predictors of MACE in patients with AMI [33,43,44].
Several previous studies have demonstrated that higher long-term mortality was observed in the NSTEMI group than in the STEMI group, regardless of the type of death [7,9,10,31,45]. Consistent with the results of previous studies [7,9,10,31,45], in all the patients in our study, the rates of MACE, all-cause death, CD, and non-CD were significantly higher in the NSTEMI group than in the STEMI group. However, the higher non-CD rate in the NSTEMI group was more evident in the DM group, while the higher CD rate in the NSTEMI group was more evident in the non-DM group. Hence, we believe that strategies for reducing the non-CD rate in patients with DM and those for reducing the CD rate in patients without DM could be beneficial for the NSTEMI group after successful PCI with new-generation DES. Although the sample size of the study population was too small to demonstrate meaningful results, more than 50 high-volume universities and community hospitals with facilities for primary PCI and on-site cardiac surgery participated in this study. Regarding the relatively higher incidence of multiple organ failure and CVA in the DM and NSTEMI groups than in the non-DM and STEMI groups (Supplementary Material 4), more well-established and regular follow-ups [9] as well as more focused and diverse secondary prevention therapies [31], including those involving lipid-lowering agents (Supplementary Materials 5 and 6), are required to reduce the occurrence of non-CD in patients with NSTEMI. Since this study enrolled patients (from the registry) from 2005, many patients with diabetes did not experience the benefits of newer therapies (such as sodium-glucose cotransporter 2 inhibitor, glucagon-like Peptide-1 receptor agonist). These medications have been shown to reduce the risk of cardiovascular events when compared with that of the controls [46,47]. Therefore, recent guidelines support the incorporation of these newer agents with cardiovascular benefits into routine clinical practice and screening of patients who are at a high risk of cardiovascular disease. Unfortunately, because we could not obtain information about the various recently developed antidiabetic agents that had been prescribed from the KAMIR registry, we could not provide information concerning the effects of these drugs on the long-term clinical outcomes in this study. Finally, we believe that the results of this comparative study could provide interventional cardiologists with meaningful information regarding treatment strategies for patients with two different types of AMI according to the presence or absence of DM.
This study has several limitations. First, because we used registry data, there may have been some under-reported or missing data. Second, this study was based on discharge medications because we could not precisely determine the participants’ adherence or non-adherence to their antidiabetic drugs during the 2-year follow-up period. Third, although the interval from symptom onset to PCI was an important determinant of major clinical outcomes, this variable included many missing values in the registry data; therefore, we could not include this variable in our study. Fourth, although we performed multivariable- and PS-adjusted analyses to strengthen our results, variables not included in the KAMIR may have affected the study outcomes. Fifth, it is not certain that our population had 100% T2DM based only on the age at which diabetes was discovered. Occasionally, type 1 DM occurs in individuals aged over 30 years [48]. Moreover, there were some missing values concerning patient-reported history, such as the presence or absence of a history of ketoacidosis and other medical records indicating T2DM in the KAMIR data, owing to the registry-based nature of this study. These factors may be considered as critical limitations of this study. Sixth, this retrospective study was a long-term (November 2005 to June 2015) study of patients with AMI, which could have affected clinical outcomes. Finally, the 2-year follow-up period in this study was relatively short and may have been inadequate for determining long-term major clinical outcomes.
- English language needs to be revised.
Answer: We sincerely thank you for reviewer’s recommendation. We totally agree with reviewer’s opinion. According to reviewer’s recommendation, we made English revisions with the help of a professional English editing company as follows and we attached a “Track change mode” file in the revision files.
- Conclusion Section: This paragraph required a general revision to eliminate redundant sentences and to add some "take-home message".
Answer: We sincerely thank you for reviewer’s valuable comments and recommendations. According to reviewer’s recommendation, we revised the “Conclusion” section as follows,
Before,
- Conclusion
In conclusion, in this retrospective study, the long-term mortality (all-cause death, CD, and non-CD) was higher in the NSTEMI group than in the STEMI group after new-generation DES implantation. However, the higher non-CD rate in the NSTEMI group was more evident in the DM group, and the higher CD rate in the NSTEMI group was more evident in the non-DM group. Hence, our results suggest that strategies for reducing the non-CD rate in patients with DM and those for reducing the CD rate in patients without DM could be beneficial in patients with NSTEMI. Further studies are required to evaluate the long-term clinical outcomes between the NSTEMI and STEMI groups more precisely.
After,
- Conclusion
In this retrospective study, patients with NSTEMI had a significantly higher 2-year mortality rate than those with STEMI did. Furthermore, strategies to reduce the non-CD rate in patients with DM and the CD rate in non-DM patients could be beneficial for those with NSTEMI. Hence, more regular follow-up and focused and diverse secondary prevention therapies to reduce the incidence of multiple organ failure and CVA are required in patients with DM and NSTEMI.
Dear respected professor reviewer 2,
We sincerely thank you for providing us an opportunity for resubmission. We also sincerely thank you for your thoughtful evaluation of our work and the valuable suggestions that helped us to improve the manuscript. We faithfully revised our paper according to your comments and recommendations as possible as.
Thank you for your consideration and we look forward to hearing from you.
Sincerely,
Seung-Woon Rha, MD., Ph.D., FACC, FAHA, FESC, FSCAI, FAPSIC.
Yong Hoon Kim, M.D., Ph.D.
